# Towards Convergence Rate Analysis of Random Forests for Classification

**Wei  Gao**         **Zhi-Hua  Zhou**
National Key Laboratory for Novel Software Technology
Nanjing University, Nanjing, 210023, China
`{gaow, zhouzh}@lamda.nju.edu.cn`

## Abstract

Random forests have been one of the successful ensemble algorithms in machine learning. The basic idea is to construct a large number of random trees individually and make prediction based on an average of their predictions. The great successes have attracted much attention on the consistency of random forests, mostly focusing on regression. This work takes one step towards convergence rates of random forests for classification. We present the first finite-sample rate $O(n^{-1/(8d+2)})$ on the convergence of pure random forests for classification, which can be improved to be of $O(n^{-1/(3.87d+2)})$ by considering the midpoint splitting mechanism. We introduce another variant of random forests, which follow Breiman's original random forests but with different mechanisms on splitting dimensions and positions. We get a convergence rate $O(n^{-1/(d+2)}(\ln n)^{1/(d+2)})$ for the variant of random forests, which reaches the minimax rate, except for a factor $(\ln n)^{1/(d+2)}$, of the optimal plug-in classifier under the $L$-Lipschitz assumption. We achieve tighter convergence rate $O(\sqrt{\ln n/n})$ under proper assumptions over structural data.

## 1   Introduction

From the pioneer work [12], random forests have been recognized as one of the successful algorithms for classification and regression, which construct a large number of random trees individually and make prediction based on an average of their predictions. This idea is partly motivated from geometric feature selection [2], random subspace [29], random split selection [23] and earlier ensemble decision trees [32]. Random forests make good performance in empirical studies [10, 12, 24, 48], and have been involved in diverse real applications such as ecology [18], computational biology [41], objection recognition [47], remote sensing [7], computer vision [16], etc. Numerous variants have been developed to improve performance and reduce computational costs [4, 6, 19, 27, 33, 34, 38, 43, 52, 56]. For an overview of random forests, we refer readers to the works of [10, 17, 26].

Empirical successes have attracted much attention on theoretical explorations of random forests. Breiman [12] presented the generalization bounds for random forests based on the correlation and strength of individual random trees, followed by consistency analysis of a simple model of random forests [13]. Lin and Jeon [35] established a connection between random forests and adaptive nearest neighbors, and Meinshausen [37] studied consistency of random forests for regression in the context of conditional quantile predictions. The consistency results place random forests in a favored category of ensemble algorithms [8, 9, 40, 44, 45, 51]. Denil et al. [20] narrowed the gap between theory and practice of random forests for regression, and Goetz et al. [28] proposed active learning algorithm for non-parametric regression using random forests. Li et al. [34] derived non-asymptotic bounds on the expected bias of MDI importance for random forests, along with variable importance [30, 36]. Tang et al. [50] discussed when random forests fail and examined the influences of parameters over performance. Most previous theoretical studies focus on random forests for regression.

For classification, Biau et al. [9] took a crucial milestone on the consistency of randomized ensemble classifiers, and Denil et al. [19] showed the first consistency of online random forests. For a fuller understanding, however, it is necessary to take one further step on the convergence rates of random forests for classification, which would be beneficial to design better random forests, and comprehend the effects of different splitting mechanisms during the constructions of random forests.

In this work, we take one step towards convergence rates of random forests for classification, and the main contributions can be summarized as follows:

- We present the first finite-sample rate on the convergence of pure random forests proposed originally by Breiman [11], that is, a convergence rate $O(n^{-1/(8d+2)})$ is derived by selecting leaves parameter $k = O(n^{4d/(4d+1)})$, where $n$ and $d$ denote the size of training data and dimension, respectively. This rate can be further improved to be of $O(n^{-1/(3.87d+2)})$ if we instead split a leaf along the dimension at the midpoint of the chosen side.

- We introduce another simplified variant of random forests, which follow Breiman's original random forests [12] but with different mechanisms on splitting dimensions and positions. We derive a convergence rate $O(n^{-1/(d+2)}(\ln n)^{1/(d+2)})$ for the simplified random forests, which reaches the minimax rate, except for a factor $(\ln n)^{1/(d+2)}$, of the optimal plug-in classifiers under the $L$-Lipschitz assumption. We finally achieve tighter convergence rate $O(\sqrt{\ln n/n})$ based on proper assumptions over structural data, which may shed insights to random forests by correlating randomization process with data-dependent tree structure.

- In addition, we establish a relationship for the convergence rates between random forests and individual random trees, and make a better estimate on the height of random trees than was previously known.

The rest of this work is organized as follows: Section 2 shows the convergence rate between random forests and individual random trees. Section 3 presents the convergence rates of pure random forests. Section 4 provides the convergence rates of the simplified variant of Breiman's original random forests. Section 5 introduces related work. Section 6 concludes with future work. Some proofs for theorems and lemmas are given in the supplementary material due to the page limitation.

## 2 Convergence Rates between Random Forests and Random Trees

Let $\mathcal{X} = [0,1]^d$ and $\mathcal{Y} = \{0,1\}$ denote the instance and label space, respectively. Suppose that $\mathcal{D}$ is an (unknown) underlying distribution over space $\mathcal{X} \times \mathcal{Y}$. Let $\mathcal{D}_{\mathcal{X}}$ be the marginal distribution over the instance space $\mathcal{X}$, and denote by

$$\eta(\boldsymbol{x}) = \Pr[y = +1|\boldsymbol{x}]$$

the conditional probability of positive instance with respect to distribution $\mathcal{D}$. In this work, we assume that conditional probability $\eta(\boldsymbol{x})$ is $L$-Lipschitz for some constant $L > 0$, i.e., for every $\boldsymbol{x}, \boldsymbol{x}' \in \mathcal{X}$,

$$|\eta(\boldsymbol{x}) - \eta(\boldsymbol{x}')| \leq L\|\boldsymbol{x} - \boldsymbol{x}'\| .$$

This assumption has been taken in random forests for regression [8, 40] and binary classification [15, 46]. Intuitively, it implies that two instances are likely to have similar labels for smaller distance. Given a hypothesis $h\colon \mathcal{X} \to \mathcal{Y}$, we define the classification error over distribution $\mathcal{D}$ as

$$R_{\mathcal{D}}(h) = \Pr_{(\boldsymbol{x},y)\sim\mathcal{D}}[h(\boldsymbol{x}) \neq y] = E_{(\boldsymbol{x},y)\sim\mathcal{D}}[\mathbb{I}[h(\boldsymbol{x}) \neq y]] .$$

Here, $\mathbb{I}[\cdot]$ denotes the indicator function, which returns 1 if the argument is true and 0 otherwise. It is well-known [22, 46] that the optimal Bayes' error (i.e., the minimum of classification error) and the Bayes' classifier can be given by

$$R_{\mathcal{D}}^* = E_{\boldsymbol{x}}[\min\{\eta(\boldsymbol{x}), 1 - \eta(\boldsymbol{x})\}] \quad \text{and} \quad h_{\mathcal{D}}^*(\boldsymbol{x}) = \mathbb{I}[\eta(\boldsymbol{x}) \geq 1/2], \quad \text{respectively.}$$

Notice that distribution $\mathcal{D}$ is unknown in practice, and what we observe is a training data

$$S_n = \{(\boldsymbol{x}_1, y_1), (\boldsymbol{x}_2, y_2), \ldots, (\boldsymbol{x}_n, y_n)\} ,$$

where each example is drawn independently and identically (i.i.d.) from distribution $\mathcal{D}$. Our goal is to learn a classifier $h_n$ with smaller classification error from the training data $S_n$. As the training

data size $n$ increases, we get a sequence of classifiers $h_1, h_2, \cdots, h_n, \cdots$. A sequence of classifiers $\{h_n\}_{n=1}^\infty$ is said to be *consistent* if $E_{S_n}[R_\mathcal{D}(h_n)] \to R_\mathcal{D}^*$ as $n \to \infty$.

Random forests classifier $f_m(\boldsymbol{x})$ takes a majority vote over $m$ individual randomized trees $f_{S_n, \Theta_1}(\boldsymbol{x})$, $f_{S_n, \Theta_2}(\boldsymbol{x}), \ldots, f_{S_n, \Theta_m}(\boldsymbol{x})$, that is,

$$f_m(\boldsymbol{x}) = \mathbb{I}\left[ \sum_{i=1}^m f_{S_n, \Theta_i}(\boldsymbol{x}) \geq \frac{m}{2} \right]. \tag{1}$$

Here, the random vectors $\Theta_1, \Theta_2, \ldots, \Theta_m$ are distributed identically and independently, and characterize the mechanisms of random selections of splitting leaves, dimensions, and positions during the construction of randomized trees. The random vectors $\Theta_1, \Theta_2, \ldots, \Theta_m$ will be specified according to different random forests in the subsequent section.

We first present the following relationship of convergence rate between random forests classifier and individual random tree classifier, and the detailed proof is given in Appendix A.

**Lemma 1** *Let $f_m(\boldsymbol{x})$ be the random forests classifier given by Eqn.* (1)*, and $f_{S_n, \Theta}(\boldsymbol{x})$ denotes a classifier of individual tree with respect to random vector $\Theta$. We have*

$$E_{\Theta_1, \ldots, \Theta_m}[R_\mathcal{D}(f_m(\boldsymbol{x}))] - R_\mathcal{D}^* \leq 2(E_\Theta[R_\mathcal{D}(f_{S_n, \Theta}(\boldsymbol{x}))] - R^*) .$$

From this lemma, the convergence rate of random forests classifier $f_m(\boldsymbol{x})$ is no more than twice that of individual random tree classifier $f_{S_n, \Theta}(\boldsymbol{x})$; therefore, the consistency of random forests can be derived from the consistency of individual random tree. A relevant result that the consistency of a random classifier is preserved by averaging [9, Proposition 1], while Lemma 1 is easier to obtain the convergence rates. In addition, the convergence rate of random forests is obtained from the expectation of convergence rates of individual trees, which can be viewed as the average of convergence rate of all of individual random trees.

It is necessary to introduce some notations used in this work. Write $[d] = \{1, 2, \ldots, d\}$ for some integer $d > 0$. We denote by $\mathcal{B}(p)$ a Bernoulli distribution with parameter $p \in [0, 1]$, and let $\mathcal{U}(a, b)$ denote a uniform distribution over the interval $[a, b]$. We further represent $\xi \sim \mathcal{B}(p)$ and $\xi \sim \mathcal{U}(a, b)$ that a random variable $\xi$ is chosen according to Bernoulli distribution $\mathcal{B}(p)$ and uniform distribution $\mathcal{U}(a, b)$, respectively. Denote by $e = 2.718...$ the Euler's constant. For positive $f(n)$ and $g(n)$, we write $f(n) = O(g(n))$ if $g(n)/f(n) \to c$ for some constant $c \in (0, +\infty)$ as $n \to \infty$.

## 3 Convergence Rates of the Pure Random Forests for Classification

We begin with the *pure random forests*, which were originally proposed by Breiman [11]. Genuer [25] studied the variance reductions of pure random forests for regression, and Arlot and Genuer [3] further presented its bias-variance analysis. For classification, Biau et al. [9] made an important milestone on the consistency of pure random forests. In this work, we take one further step on the convergence rate of pure random forests for classification.

Formally, a pure random tree can be constructed as follows. Each node is associated with a rectangular cell, and all leaves (external nodes) constitute a partition of $[0, 1]^d$ at each iteration of tree construction. The root of random partition is $[0, 1]^d$ itself. The following procedure is repeated $k - 1$ iterations for some pre-defined $k \geq 2$ in advance, and hence the output random tree has $k$ leaves.

- A split leaf is selected at random, uniformly over all leaves at the current iteration.
- Once the leaf is selected, a split dimension is selected at random, uniformly over $[d]$.
- The leaf is split along the split dimension at random, uniformly over the chosen side.

A pure random tree classifier $f_{S_n, \Theta}(\boldsymbol{x})$ takes a majority vote over labels $y_i$ whose corresponding instances $\boldsymbol{x}_i$ belong to the same cell of random partition as instance $\boldsymbol{x}$. The main difference, between pure random tree and Breiman's original random tree [12], is that recursive cell splits are irrelevant to label information, and the growth of individual random tree is independent of training sample.

Given $m$ individual pure random trees $f_{S_n, \Theta_1}(\boldsymbol{x})$, $f_{S_n, \Theta_2}(\boldsymbol{x}), \ldots, f_{S_n, \Theta_m}(\boldsymbol{x})$, the random forests classifier takes a majority vote over those random trees, that is, $f_m(\boldsymbol{x}) = \mathbb{I}[\sum_{i=1}^m f_{S_n, \Theta_i}(\boldsymbol{x}) \geq m/2]$. We now present the convergence rates of pure random forests for classification.

**Theorem 1** *Let $f_m(\boldsymbol{x})$ be the random forests classifier by applying pure random tree to training data $S_n$ of $k$ leaves $(k \geq 2)$. For L-Lipschitz conditional probability $\eta(\boldsymbol{x})$, we have*

$$R_{\mathcal{D}}^* \leq E_{S_n, \Theta_1, \ldots, \Theta_m}[R_{\mathcal{D}}(f_m)] \leq R_{\mathcal{D}}^* + \frac{4\sqrt{2eL d^{3/2}}}{k^{1/8d}} + 2\sqrt{\frac{k}{n}} + \frac{6k}{n} \ .$$

Based on this theorem, we obtain a convergence rate $O(n^{-1/(8d+2)})$ of pure random forests for classification, by selecting leaves parameter $k = O(n^{4d/(4d+1)})$. To the best of our knowledge, this presents the first finite-sample converge rate of pure random forests for classification. Also, it is easy to observe that

$$E_{S_n, \Theta_1, \ldots, \Theta_m}[R_{\mathcal{D}}(f_m)] \to R_{\mathcal{D}}^* \quad \text{as} \quad k \to +\infty \quad \text{and} \quad k/n \to 0,$$

which recovers the consistency result of random forests for classification [9, Theorem 2].

Before the proof of Theorem 1, we go into the details of randomness $\Theta$ on the construction of pure random forests. Given a pure random tree, we associate $k$ leaves with $k$ disjoint rectangular cells $C_1, C_2, \ldots, C_k$, constituting a partition of instance space $\mathcal{X} = [0,1]^d$. Given an instance $\boldsymbol{x} \in \mathcal{X}$, let $C(\boldsymbol{x})$ denote the rectangular cell of the random tree, that contains the instance $\boldsymbol{x}$.

Given an instance $\boldsymbol{x} \in \mathcal{X}$, we introduce $k-1$ Bernoulli random variables $X_1, X_2, \cdots, X_{k-1}$ to characterize the random events that the node containing instance $\boldsymbol{x}$ was selected for splitting in the construction of random tree. Specially, the event $X_i = 1$ implies that the node containing $\boldsymbol{x}$ is selected for splitting in the $i$-th iteration of random tree construction; otherwise, $X_i = 0$. It follows that $X_i \sim \mathcal{B}(1/i)$, because there are $i$ leaves for selection with identical probability in the $i$-th iteration of random tree construction.

Let $h(C(\boldsymbol{x}))$ denote the height of the rectangular cell $C(\boldsymbol{x})$, i.e., the splitting times of $C(\boldsymbol{x})$ during the construction of random tree. It is easy to obtain

$$h(C(\boldsymbol{x})) = \sum_{i=1}^{k-1} X_i \ .$$

We further present upper and lower bounds on $h(C(\boldsymbol{x}))$ in expectation and in probability as follows:

**Lemma 2** *Let $X_1, X_2, \ldots, X_{k-1}$ be $k-1$ random variables such that $X_i \sim \mathcal{B}(1/i)$ for $i \in [k-1]$. For an instance $\boldsymbol{x} \in \mathcal{X}$, we have*

$$\ln(k) \leq E_{X_1, X_2, \ldots, X_{k-1}}[h(C(\boldsymbol{x}))] \leq 1 + \ln(k-1) \ ,$$

*and we also have, for any $\epsilon \in (0,1)$,*

$$\Pr\nolimits_{X_1, X_2, \ldots, X_{k-1}}[h(C(\boldsymbol{x})) \leq (1-\epsilon)\ln k] \leq k^{-\epsilon^2/2} \ ,$$
$$\Pr\nolimits_{X_1, X_2, \ldots, X_{k-1}}[h(C(\boldsymbol{x})) \geq (1+\epsilon)(1+\ln(k-1))] \leq k^{-\epsilon^2/2} \ .$$

We have $h(C(\boldsymbol{x})) = O(\log k)$ with large probability, especially for large $k$. Lemma 2 improves the previous work [9] on the bounds of $h(C(\boldsymbol{x}))$, where the saturation level is considered in random binary search tree [21, 42], and their bounds can be rewritten (with our notation) as follows:

$$\Pr[h(C(\boldsymbol{x})) < (c^* - \epsilon)\ln k] \leq O(\log(k)k^{(c^*-\epsilon)\ln(2e/(c^*-\epsilon))-1}) \ .$$

Here, $c^* = 0.3733\ldots$ is the unique solution of $c\ln(2e/c) = 1$ $(c < 1)$ and $\epsilon < c^*$. As can be seen, Lemma 2 makes better estimations of $h(C(\boldsymbol{x}))$ with larger probability. The detailed proof of Lemma 2 is presented in Appendix B.

Given a cell $C(\boldsymbol{x})$, we define its diameter as $\nu(C(\boldsymbol{x})) = \max_{\boldsymbol{x}, \boldsymbol{x}' \in C(\boldsymbol{x})}\{\|\boldsymbol{x} - \boldsymbol{x}'\|\}$. Then, we can bound $\nu(C(\boldsymbol{x}))$ in probability as follows:

**Lemma 3** *For integer $k \geq 2$, real $\epsilon > -1$ and instance $\boldsymbol{x} \in \mathcal{X}$, we have*

$$\Pr\left[\nu[C(\boldsymbol{x})] \geq (1+\epsilon)\frac{\sqrt{d}}{k^{1/8d}}\right] \leq \frac{ed}{(1+\epsilon)k^{1/8d}} \ ,$$

*where the probability takes over the random selections of splitting leaves, dimensions and positions.*

This lemma shows that, for every instance $\boldsymbol{x} \in \mathcal{X}$, the diameter of rectangle cell of $C(\boldsymbol{x})$ can be upper bounded by $(1+\epsilon)\sqrt{d}/k^{1/8d}$ with probability at least $1 - ed/(1+\epsilon)k^{1/8d}$. We also have $\nu(C(\boldsymbol{x})) \to 0$ in probability as $k \to +\infty$. For simplicity, we do not formalize the random selections of splitting leaves, dimensions and positions in Lemma 3, while the detailed formalization and proof are presented in Appendix C.

Recall that there are $k$ disjoint rectangular cells $C_1, C_2, \ldots, C_k$ during the construction of pure random tree with $k-1$ iterations. We present the following lemma to bound the classification error over each rectangular cell, and the detailed proof is given in Appendix D.

**Lemma 4** *Let $C_1, C_2, \ldots, C_k$ be the $k$ disjoint rectangular cells associating with the leaves of randomized tree, and $f_{\Theta, S_n}(\boldsymbol{x})$ denotes the classifier generated by random tree. For L-Lipschitz conditional probability $\eta(\boldsymbol{x})$ and for every $i \in [k]$, we have*

$$\Pr_{S_n, (\boldsymbol{x}, y)}[f_{\Theta, S_n}(\boldsymbol{x}) \neq y | \boldsymbol{x} \in C_i] \Pr[\boldsymbol{x} \in C_i] \leq 2L\nu(C_i) \Pr[\boldsymbol{x} \in C_i]$$

$$+ E_{\boldsymbol{x}}[\min\{\eta(\boldsymbol{x}), 1 - \eta(\boldsymbol{x})\} | \boldsymbol{x} \in C_i] \Pr[\boldsymbol{x} \in C_i] + \sqrt{\Pr[\boldsymbol{x} \in C_i]/n} + 3/n \ .$$

Based on the previous lemmas, we now present the detailed proof of Theorem 1 as follow:

**Proof of Theorem 1**. We first derive the convergence rate of individual random tree classifier $f_{S_n, \Theta}(\boldsymbol{x})$, and then complete the proof by combining with Lemma 1. We have

$$R_{\mathcal{D}}(f_{S_n, \Theta}) = \Pr_{(\boldsymbol{x}, y) \sim \mathcal{D}}[f_{\Theta, S_n}(\boldsymbol{x}) \neq y] = E_{\boldsymbol{x} \sim \mathcal{D}_{\mathcal{X}}}\left[\Pr_{y \sim \mathcal{B}(\eta(\boldsymbol{x}))}[f_{\Theta, S_n}(\boldsymbol{x}) \neq y]\right] \ .$$

For random tree classifier $f_{\Theta, S_n}(\boldsymbol{x})$, we associate a set as follows:

$$\Lambda = \left\{\boldsymbol{x} \in \mathcal{X} : \nu(C(\boldsymbol{x})) \geq (1+\epsilon)\sqrt{d}/k^{1/8d}\right\} \ , \tag{2}$$

where $\nu(C(\boldsymbol{x}))$ denotes the diameter of rectangle cell $C(\boldsymbol{x})$. It follows that

$$R_{\mathcal{D}}(f_{S_n, \Theta}) = E_{\boldsymbol{x} \sim \mathcal{D}_{\mathcal{X}}}\left[\Pr_{y \sim \mathcal{B}(\eta(\boldsymbol{x}))}[f_{\Theta, S_n}(\boldsymbol{x}) \neq y](\mathbb{I}[\boldsymbol{x} \in \Lambda] + \mathbb{I}[\boldsymbol{x} \notin \Lambda])\right]$$

$$\leq E_{\boldsymbol{x} \sim \mathcal{D}_{\mathcal{X}}}[\mathbb{I}[\boldsymbol{x} \in \Lambda]] + E_{\boldsymbol{x} \sim \mathcal{D}_{\mathcal{X}}}\left[\Pr_{y \sim \mathcal{B}(\eta(\boldsymbol{x}))}[f_{\Theta, S_n}(\boldsymbol{x}) \neq y]\mathbb{I}[\boldsymbol{x} \notin \Lambda]\right] \ . \tag{3}$$

Notice that $C_1, C_2, \ldots, C_k$ is a partition of the instance space $\mathcal{X}$ from the construction of random tree. Based on the law of total probability, we have

$$E_{\boldsymbol{x} \sim \mathcal{D}_{\mathcal{X}}}\left[\Pr_{y \sim \mathcal{B}(\eta(\boldsymbol{x}))}[f_{\Theta, S_n}(\boldsymbol{x}) \neq y]\mathbb{I}[\boldsymbol{x} \notin \Lambda]\right]$$

$$= \sum_{i=1}^{k} \Pr[f_{\Theta, S_n}(\boldsymbol{x}) \neq y | \boldsymbol{x} \in C_i] \Pr[\boldsymbol{x} \in C_i]\mathbb{I}[C_i \not\subseteq \Lambda] \ ,$$

where we use the fact $C(\boldsymbol{x}) = C_i$ for every $\boldsymbol{x} \in C_i$. By combining with Eqns. (2) and (3), we have

$$E_{S_n, \Theta}[R_{\mathcal{D}}(f_{S_n, \Theta})] \leq E_{\boldsymbol{x} \sim \mathcal{D}_{\mathcal{X}}}\left[\Pr_{S_n, \Theta}\left[\nu[C(\boldsymbol{x})] \geq (1+\epsilon)\sqrt{d}/k^{1/8d}\right]\right] \tag{4}$$

$$+ E_{\Theta}\left[\sum_{i=1}^{k} E_{S_n}[\Pr[f_{\Theta, S_n}(\boldsymbol{x}) \neq y | \boldsymbol{x} \in C_i]] \Pr[\boldsymbol{x} \in C_i]\mathbb{I}[C_i \not\subseteq \Lambda]\right] \ . \tag{5}$$

From Lemma 3, Eqn. (4) can be further upper bounded by

$$E_{\boldsymbol{x} \sim \mathcal{D}_{\mathcal{X}}}\left[\Pr_{S_n, \Theta}\left[\nu[C(\boldsymbol{x})] \geq (1+\epsilon)\sqrt{d}/k^{1/8d}\right]\right] \leq \frac{ed}{(1+\epsilon)k^{1/8d}} \ . \tag{6}$$

Based on Lemma 4 and Eqn. (2), we can bound Eqn. (5) as follows

$$\sum_{i=1}^{k} E_{S_n}[\Pr[f_{\Theta, S_n}(\boldsymbol{x}) \neq y | \boldsymbol{x} \in C_i]] \Pr[\boldsymbol{x} \in C_i]\mathbb{I}[C_i \not\subseteq \Lambda]$$

$$\leq R_{\mathcal{D}}^* + \frac{2(1+\epsilon)L\sqrt{d}}{k^{1/8d}} + \sum_{i=1}^{k} \sqrt{\frac{\Pr[C_i]}{n}} + \frac{3k}{n} \ , \tag{7}$$

where we use the law of total expectation and $R_{\mathcal{D}}^* = E_{\boldsymbol{x} \sim \mathcal{D}_{\mathcal{X}}}[\min\{\eta(\boldsymbol{x}), 1 - \eta(\boldsymbol{x})\}]$. By Jensen's inequality, we have $(EX)^2 \le E[X^2]$, and this gives

$$\left( \frac{1}{k} \sum_{i=1}^{k} \sqrt{\Pr[C_i]} \right)^2 \le \frac{1}{k} \sum_{i=1}^{k} \Pr[C_i] = \frac{1}{k} .$$

It follows that, by combining with Eqns. (4)-(7),

$$E_{S_n, \Theta}[R_{\mathcal{D}}(f_{S_n, \Theta})] \le R_{\mathcal{D}}^* + \frac{ed}{(1+\epsilon)k^{1/8d}} + \frac{2(1+\epsilon)L\sqrt{d}}{k^{1/8d}} + \sqrt{\frac{k}{n}} + \frac{3k}{n} .$$

We have, by setting $\epsilon = \sqrt{e\sqrt{d}/2L}$ and algebra calculations,

$$E_{S_n, \Theta}[R_{\mathcal{D}}(f_{S_n, \Theta})] \le R_{\mathcal{D}}^* + \frac{2\sqrt{2eL d^{3/2}}}{k^{1/8d}} + \sqrt{\frac{k}{n}} + \frac{3k}{n} ,$$

which completes the proof by combining with Lemma 1. ∎

We further study the effects of different splitting mechanisms during the construction of random forests. For example, how about the convergence rates for different selections of splitting leaves, dimensions and positions? Here, we consider *pure random forests with midpoint splits*, where midpoint splits have been well-studied for random forests in regression [3, 8, 31]. Formally, a pure random tree with midpoint splits can be constructed as follows. The root of random partition is $[0, 1]^d$ itself. The following procedure is repeated $k - 1$ iterations for some pre-defined $k \ge 2$ in advance.

- A split leaf is selected at random, uniformly over all leaves at the current iteration.
- Once the leaf is selected, a split dimension is selected at random, uniformly over $[d]$.
- The leaf is split along the split dimension at the midpoint of the chosen side.

Given individual random tree classifiers $f_{S_n, \Theta_1}(\boldsymbol{x}), f_{S_n, \Theta_2}(\boldsymbol{x}), \dots, f_{S_n, \Theta_m}(\boldsymbol{x})$, the random forests classifier takes a majority vote over $m$ random trees. We present a convergence rate of pure random forests with midpoint splits for classification as follows:

**Theorem 2** *Let $f_m(\boldsymbol{x})$ be the random forests classifier by applying pure random tree with midpoint splits to training data $S_n$ of $k$ leaves ($k \ge 2$). For $L$-Lipschitz conditional probability $\eta(\boldsymbol{x})$, we have*

$$R_{\mathcal{D}}^* \le E_{S_n, \Theta_1, \dots, \Theta_m}[R_{\mathcal{D}}(f_m)] \le R_{\mathcal{D}}^* + \frac{8L^{3/5}d^{7/10}}{k^{1/3.87d}} + 2\sqrt{\frac{k}{n}} + \frac{6k}{n} .$$

Based on this theorem, we get a convergence rate $O(n^{-1/(3.87d+2)})$ of pure random forests with midpoint splits for classification, by selecting leaves parameter $k = O(n^{3.87d/(3.87d+2)})$. As can be seen, we achieve better convergence rate by instead considering the midpoint splitting mechanism during the construction of pure random forests, and an intuitive explanation is that midpoint splits yield smaller rectangle cells. The detailed proof of Theorem 2 is presented in Appendix E.

## 4 Convergence Rates of the Simplified Random Forests for Classification

In this section, we present the convergence analysis towards Breiman's original random forests [12] for classification. We follow the procedures of Breiman's random forests, but with different mechanisms on the selections of splitting dimensions and positions due to technical analysis challenges. Algorithm 1 gives a detailed description of the simplified variant of random forests.

We introduce a structural list $\mathcal{P}$ to store leaves (or rectangle cells) for further splitting in Algorithm 1, which aims to keep the leaves split in successive layer. Such mechanism is essentially the same as that of random forests for regression [45]. At each iteration, the first leaf (or rectangle cell) is selected and removed from $\mathcal{P}$, and it will not be split if all training examples have the same label in the leaf (including less than one example in the leaf). For a split leaf, we select a dimension at

---

**Algorithm 1** A simplified variant of Breiman's original random tree [12]

---

**Input**: Training sample $S_n$ and leaves parameter $k$.
**Output**: A random tree
**Initialize**: Set $\mathcal{P} = \{[0,1]^d\}$ and $n_{\text{leaf}} = 1$.

 1: **while** $n_{\text{leaf}} < k$ **and** $\mathcal{P}$ is not empty **do**
 2:     Let $C$ be the first rectangle cell in $\mathcal{P}$, and remove it from $\mathcal{P}$.
 3:     **if** All training examples in $C$ have the same label (including less than one example) **then**
 4:         Do nothing and the cell $C$ will not be split any more.
 5:     **else**
 6:         Select a dimension $Y$ at random, uniformly over dimensions along which the side length is maximal in the cell $C$.
 7:         Split cell $C$ along $Y$ at the midpoint of the chosen side, called $C_L, C_R$ two resulting cells.
 8:         Update $\mathcal{P}$ by appending $C_L$ and $C_R$, and $n_{\text{leaf}} \leftarrow n_{\text{leaf}} + 1$.
 9:     **end if**
10: **end while**

---

random, uniformly over dimensions along which the side length is maximal in the leaf, and then split the leaf along the dimension at the midpoint of the chosen side. We finally update list $\mathcal{P}$ by appending two resulting leaves.

A leaf (rectangle cell) will not be split in Algorithm 1 if all training examples have the same label in this leaf. Such stopping-splitting criterion is different from pure random forests [11] and Mondrian forests [33, 40], where the growth of individual random tree is independent of training sample. In addition, it is also different from random forests regression [45], where a leaf will not be split only when the leaf has exactly one training example.

Given $m$ individual random tree classifiers $f_{S_n,\Theta_1}(\boldsymbol{x})$, $f_{S_n,\Theta_2}(\boldsymbol{x}), \ldots, f_{S_n,\Theta_m}(\boldsymbol{x})$ according to Algorithm 1, the random forests classifier takes a majority vote over $m$ random trees, that is, $f_m(\boldsymbol{x}) = \mathbb{I}\left[\sum_{i=1}^m f_{S_n,\Theta_i}(\boldsymbol{x}) \geq m/2\right]$. We present a convergence rate of the simplified variant of random forests for classification as follows:

**Theorem 3** *For $k \geq 2$ and $n \geq 4$, let $f_m(\boldsymbol{x})$ be the random forests classifier by applying Algorithm 1 to training data $S_n$ of $k$ leaves. For $L$-Lipschitz conditional probability $\eta(\boldsymbol{x})$, we have*

$$R_{\mathcal{D}}^* \leq E_{S_n,\Theta_1,\ldots,\Theta_m}[R_{\mathcal{D}}(f_m)] \leq R_{\mathcal{D}}^* + 4\sqrt{\frac{k \ln n}{n}} + 2\sqrt[4]{\frac{4k^3 \ln n}{n^3}} + \frac{12k}{n} + 4\sqrt{\frac{k}{n}} + \frac{4L\sqrt{d}}{k^{1/d}} \ .$$

We obtain a convergence rate $O(n^{-1/(d+2)}(\ln n)^{1/(d+2)})$ for random forests based on Algorithm 1, by selecting leaves parameter $k = O((n/\ln n)^{2d/(d+2)})$. This presents significantly better convergence rate than that of pure random forests due to different splitting mechanisms and stopping-splitting criteria. The detailed proof of Theorem 3 is given in Appendix F.

Under the $L$-Lipschitz assumption, it is well-known [5, 54] that the minimax rate is of $O(n^{-1/(d+2)})$ for the optimal plug-in classifiers $f(\boldsymbol{x}) = \mathbb{I}[\hat{\eta}(\boldsymbol{x}) \geq 1/2]$, where $\hat{\eta}(\boldsymbol{x})$ is a conditional probability estimated by learning algorithms. As can be seen, our simplified variant of random forests reaches the minimax convergence rate, except for a factor $(\ln n)^{1/(1+d)}$, as that of the optimal plug-in classifiers, despite random forests are not plug-in classifiers, since random forests take a majority vote over the predictions of individual random trees, rather than the estimation of conditional probability.

Breiman's original random forests [12] took some splitting criteria, such as information gain and entropy, to select the best-split dimension and position, which correlates the randomization process with data-dependent tree structure. Intuitively, such correlation could yield tighter convergence rates of random forests for classification, whereas this makes it quite challenging to present convergence analysis from a technical view. To date, it is still an open problem on the consistency of Breiman's original random forests [12] for classification, let alone the analysis of convergence rate.

We now make some assumptions over structural data, which could yield tighter convergence rate for the simplified variant of random forests. Suppose that there is a constant $k_0 \geq 2$, such that the output random trees from Algorithm 1 have at most $k_0$ leaves with all training examples in each leaf having the same label. Based on such assumption, we present a convergence rate of the simplified variant of random forests for classification.

**Theorem 4** *Suppose that there is a constant $k_0 \geq 2$, such that the output random trees from Algorithm 1 have at most $k_0$ leaves with all training examples having the same label in each leaf. Let $f_m(\boldsymbol{x})$ be the random forests classifier by applying Algorithm 1 to training data $S_n$. We have*

$$E_{S_n, \Theta_1, \ldots, \Theta_m}[R_{\mathcal{D}}(f_m)] \leq 4\sqrt{\frac{k_0 \ln n}{n}} + 2\sqrt[4]{\frac{4k_0^3 \ln n}{n^3}} + 2\sqrt{\frac{k_0}{n \ln n}} + \frac{6k_0}{n} \ .$$

Based on this theorem, we achieve tighter convergence rate $O(\sqrt{\ln n / n})$ of the simplified variant of random forests for classification, which is independent of dimension $d$. This theorem may show some lights on Breiman's original random forests [12] with tighter convergence rates, by correlating randomization process and data-dependent tree structure.

The assumption in Theorem 4 is relevant to algorithm, while it still holds for some irrelevant cases, for example, Algorithm 1 satisfies such assumption when the data is separable and the separable hyperplane is parallel to axis. The detailed proof of Theorem 4 is presented in Appendix G.

## 5 Related Work

For random forests, a large number of variants have been developed according to different problems and settings in the literature during the past decades. Geurts et al. [27] introduced the *extremely randomized trees* and Amaratunga et al. [1] provided the *enriched random forests* for DNA microarray data of huge features. Menze et al. [38] presented the *oblique random forests* for multivariate trees by explicitly learning the optimal split directions with linear discriminative models. Clémençon et al. [14] introduced the *ranking forests* based on aggregation and feature randomization principles for bipartite ranking. Athey et al. [4] developed a flexible and computationally efficient algorithm for the generalized random forests. A general framework is presented in [53] on various splitting criteria for random forests based on loss functions. Zhou and Feng [55, 56] proposed *gcForest* with performance highly competitive to deep neural networks. Online random forests have also been developed with strong theoretical guarantees [19, 33, 40, 49].

For regression, much attention has been paid on the $\mathbb{L}_2^2$-consistency of random forests with relevant variants [3, 8, 20, 25, 37, 45]. In particular, Scornet et al. [45] proved the first $\mathbb{L}_2^2$-consistency of Breiman's original random forests based on some assumptions such as additive regression functions and uniform distribution over instance space $\mathcal{X}$. The crucial analysis technique is the classical decomposition of variance and bias for random forests regression, whereas it is difficult to make such decomposition for random forests in classification. Moreover, the stopping-splitting criteria are different for random forests classification and regression, as shown in Algorithm 1 and work [45], respectively. We do not directly compare the convergence rates of random forests for regression and classification due to different settings and performance measures.

For classification, Biau et al. [9] made a crucial milestone on the consistency of several randomized ensemble classifiers such as pure random forests. The key technical tool is the general consistency theorem for partition classifiers [22, Theorem 6.1], that is, partition classifiers are consistent if the followings hold in probability (written with our notations),

$$\nu(C(\boldsymbol{x})) \to 0 \ \text{ and } \ |C(\boldsymbol{x}) \cap S_n| \to +\infty \ \text{ as } \ n \to +\infty \ ,$$

where $\nu(C(\boldsymbol{x}))$ denotes the diameter of the leaf or rectangular cell $C(\boldsymbol{x})$. Along this line, many random forests classifiers have been proven to be consistent such as random forests model [8], online random forests [19], online Mondrian forests [40], etc. Our work presents the convergence rates of random forests for classification based on different analysis techniques, and it is interesting to study the convergence rates of other variants of random forests along our analysis.

Mourtada et al. [40] presented the consistency of online Mondrian forests classifiers according to [22, Theorem 6.1], and derived the minimax rate $O(n^{-1/(d+2)})$ for plug-in classifiers based on the estimation of conditional probability, that is, they took an average of conditional probabilities calculated by individual Mondrian trees. This is different from random forests classifier, which takes a majority over the predictions made by individual random trees. Also, the growth of individual Mondrian tree is independent of training sample, which is different from Algorithm 1.

# 6    Conclusion

This work takes one step towards the convergence rate analysis of random forests for classification. We present the first finite-sample convergence rate $O(n^{-1/(8d+2)})$ for pure random forests, as well as a convergence rate $O(n^{-1/(d+2)}(\ln n)^{1/(d+2)})$ for the simplified variant of Breiman's original random forests [12], which reaches the minimax rate, except for a factor $(\ln n)^{1/(d+2)}$, of the optimal plug-in classifier under the $L$-Lipschitz assumption. It is still a long way to fully understand random forests and relevant mechanisms such as bootstrap sampling, data-dependence tree structure, tree pruning, etc., and we leave those to future work. In addition, it is also interesting to extend our work to multi-class learning, where the challenges lie in the theoretical analysis of predictions $f(x, y) - \max_{i \neq y} f(x, i)$ and Lipschitz assumptions over multiple class-conditional distributions.

## Broader Impact

This work presents theoretical analysis on the convergence rates of random forests in the machine learning community. This is a pure theoretical work without particular application foreseen.

## Acknowledgments and Disclosure of Funding

The authors would like to thank the reviewers for helpful comments and suggestions. This research was supported by the NSFC (61921006, 61876078), the Fundamental Research Funds for the Central Universities (14380003).

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
