[Supplementary Material]

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

$$+ E_x[\min\{\eta(x), 1 - \eta(x)\} | x \in C_i] \Pr[x \in C_i] + \sqrt{\Pr[x \in C_i]/n} + 3/n .$$

Based on the previous lemmas, we now present the detailed proof of Theorem 1 as follow:

**Proof of Theorem 1**. We first derive the convergence rate of individual random tree classifier $f_{S_n, \Theta}(x)$, and then complete the proof by combining with Lemma 1. We have

$$R_{\mathcal{D}}(f_{S_n, \Theta}) = \Pr_{(x,y) \sim \mathcal{D}}[f_{\Theta, S_n}(x) \neq y] = E_{x \sim \mathcal{D}_{\mathcal{X}}}\left[\Pr_{y \sim \mathcal{B}(\eta(x))}[f_{\Theta, S_n}(x) \neq y]\right] .$$

For random tree classifier $f_{\Theta, S_n}(x)$, we associate a set as follows:

$$\Lambda = \left\{ x \in \mathcal{X} : \nu(C(x)) \geq (1+\epsilon)\sqrt{d}/k^{1/8d} \right\} , \tag{2}$$

where $\nu(C(x))$ denotes the diameter of rectangle cell $C(x)$. It follows that

$$R_{\mathcal{D}}(f_{S_n, \Theta}) = E_{x \sim \mathcal{D}_{\mathcal{X}}}\left[\Pr_{y \sim \mathcal{B}(\eta(x))}[f_{\Theta, S_n}(x) \neq y](\mathbb{I}[x \in \Lambda] + \mathbb{I}[x \notin \Lambda])\right]$$

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

_m$ be $m$ independent random variables with $X_i \in [0, 1]$. Denote by $X = \sum_{i=1}^m X_i$ and $p = \sum_{i=1}^m E[X_i]$. We have*

$$\Pr_{X_1,\ldots,X_m} [X < (1-\delta)p] \leq \exp(-p\delta^2/2) \ .$$

This lemma is a variant of Chernoff bounds from [39].

**Lemma 6** *For any integer $k \geq 2$, we have*

$$\ln k \leq \sum_{i=1}^{k-1} \frac{1}{i} \leq 1 + \ln(k-1) \ .$$

*Proof:* For any integer $i > 0$, we have

$$\frac{1}{t} \leq \frac{1}{i} \ \text{ for } \ t \in [i, i+1] \qquad \text{and} \qquad \frac{1}{i} \leq \frac{1}{t} \ \text{ for } \ t \in [i-1, i].$$

It follows that

$$\ln k = \int_1^k \frac{1}{t} dt \leq \sum_{i=1}^{k-1} \frac{1}{i} \leq 1 + \int_1^{k-1} \frac{1}{t} dt = 1 + \ln(k-1) \ ,$$

which completes the proof. ∎

**Lemma 7** *For integers $k \geq 2$ and $d \geq 2$, we have*

$$\sum_{i=1}^{k-1} \ln\left(1 - \frac{3}{4id}\right) \geq -\frac{9 + 3\ln(k-1)}{4d} \ , \tag{8}$$

$$\sum_{i=1}^{k-1} \ln\left(1 - \frac{1}{2id}\right) \geq -\frac{3 + \ln(k-1)}{2d} \ . \tag{9}$$

*Proof:* We have

$$\sum_{i=1}^{k-1} \ln\left(1 - \frac{3}{4id}\right)$$

$$\geq \ln\left(1 - \frac{3}{4d}\right) + \int_1^{k-1} \ln\left(1 - \frac{3}{4dt}\right) dt$$

$$= \ln\left(1 - \frac{3}{4d}\right) + \left[ t \ln\left(1 - \frac{3}{4dt}\right)\right]_1^{k-1} - \int_1^{k-1} \frac{3}{4dt - 3} dt$$

$$= (k-1) \ln\left(1 - \frac{3}{4d(k-1)}\right) - \frac{3}{4d}\ln\left(k - 1 - \frac{3}{4d}\right) + \frac{3}{4d}\ln\left(1 - \frac{3}{4d}\right) \ .$$

It is easy to observe $\ln(k - 1 - 3/4d) \leq \ln(k-1)$, and

$$(k-1)\ln\left(1 - \frac{3}{4d(k-1)}\right) + \frac{3}{4d}\ln\left(1 - \frac{3}{4d}\right) \geq -\frac{3}{2d} - \frac{9}{8d^2} \geq \frac{-33}{16d} \geq \frac{-9}{4d} \ ,$$

by using $\ln(1-x) > -2x$ for $x \in [0, 1/2]$ and $d \geq 2$. Eqn. (8) holds by simple calculations.

For Eqn. (9), we similarly have

$$\sum_{i=1}^{k-1} \ln\left(1 - \frac{1}{2id}\right)$$

$$\geq \ln\left(1 - \frac{1}{2d}\right) + \int_1^{k-1} \ln\left(1 - \frac{1}{2dt}\right) dt$$

$$= \ln\left(1 - \frac{1}{2d}\right) + \left[t \ln\left(1 - \frac{1}{2dt}\right)\right]_1^{k-1} - \int_1^{k-1} \frac{1}{2dt - 1} dt$$

$$= (k-1) \ln\left(1 - \frac{1}{2d(k-1)}\right) - \frac{1}{2d} \ln\left(k - 1 - \frac{1}{2d}\right) + \frac{1}{2d} \ln\left(1 - \frac{1}{2d}\right).$$

It follows that, by using $\ln(1 - x) > -2x$ for $x \in [0, 1/2]$ and for $d \geq 2$,

$$(k-1) \ln\left(1 - \frac{1}{2d(k-1)}\right) + \frac{1}{2d} \ln\left(1 - \frac{1}{2d}\right) \geq -\frac{1}{d} - \frac{1}{2d^2} \geq -\frac{5}{4d} \geq -\frac{3}{2d}.$$

This completes the proof of Eqn. (9) by simple calculations. ∎

## A  Proof of Lemma 1

According to the definitions of $R_{\mathcal{D}}(h)$ and $R_{\mathcal{D}}^*$, we have

$$E_\Theta[R_{\mathcal{D}}(f_{S_n,\Theta})] - R_{\mathcal{D}}^*$$

$$= E_{\boldsymbol{x} \sim \mathcal{D}_{\mathcal{X}}} \left[ (1 - 2\eta(\boldsymbol{x}))\mathbb{I}[\eta(\boldsymbol{x}) < 1/2] \Pr_\Theta[f_{S_n,\Theta}(\boldsymbol{x}) = 1] \right.$$

$$\left. + (2\eta(\boldsymbol{x}) - 1)\mathbb{I}[\eta(\boldsymbol{x}) > 1/2] \Pr_\Theta[f_{S_n,\Theta}(\boldsymbol{x}) = 0] \right].$$

We similarly have

$$E_{\Theta_1,\ldots,\Theta_m}[R_{\mathcal{D}}(f_m)] - R_{\mathcal{D}}^*$$

$$= E_{\boldsymbol{x} \sim \mathcal{D}_{\mathcal{X}}} \left[ (1 - 2\eta(\boldsymbol{x}))\mathbb{I}[\eta(\boldsymbol{x}) < 1/2] \Pr_{\Theta_1,\ldots,\Theta_m}[f_m(\boldsymbol{x}) = 1] \right.$$

$$\left. + (2\eta(\boldsymbol{x}) - 1)\mathbb{I}[\eta(\boldsymbol{x}) > 1/2] \Pr_{\Theta_1,\ldots,\Theta_m}[f_m(\boldsymbol{x}) = 0] \right].$$

Given any instance $\boldsymbol{x} \in \mathcal{X}$ with $\eta(\boldsymbol{x}) < 1/2$, we have, by Markov's inequality,

$$\Pr_{\Theta_1,\ldots,\Theta_m}[f_m(\boldsymbol{x}) = 1] = \Pr_{\Theta_1,\ldots,\Theta_m}\left[ \sum_{i=1}^m f_{S_n,\Theta_i}(\boldsymbol{x}) \geq \frac{m}{2} \right]$$

$$\leq \frac{2}{m} \sum_{i=1}^m E_{\Theta_i}[f_{S_n,\Theta_i}(\boldsymbol{x})]$$

$$= 2 \Pr_\Theta[f_{S_n,\Theta}(\boldsymbol{x}) = 1],$$

from the fact $E_{\Theta_i}[f_{S_n,\Theta_i}(\boldsymbol{x})] = \Pr_{\Theta_i}[f_{S_n,\Theta_i}(\boldsymbol{x}) = 1]$ for $f_{S_n,\Theta_i}(\boldsymbol{x}) \in \{0, 1\}$.

For any instance $\boldsymbol{x} \in \mathcal{X}$ with $\eta(\boldsymbol{x}) > 1/2$, we also have

$$\Pr_{\Theta_1,\ldots,\Theta_m}[f_m(\boldsymbol{x}) = 0] = \Pr_{\Theta_1,\ldots,\Theta_m}\left[ \sum_{i=1}^m f_{S_n,\Theta_i}(\boldsymbol{x}) < \frac{m}{2} \right]$$

$$= \Pr_{\Theta_1,\ldots,\Theta_m}\left[ \sum_{i=1}^m \mathbb{I}[f_{S_n,\Theta_i}(\boldsymbol{x}) = 0] \geq \frac{m}{2} \right]$$

$$\leq \frac{2}{m} \sum_{i=1}^m E_{\Theta_i}[\mathbb{I}[f_{S_n,\Theta_i}(\boldsymbol{x}) = 0]]$$

$$= 2 \Pr_\Theta[f_{S_n,\Theta}(\boldsymbol{x}) = 0].$$

This completes the proof by combining with the trivial case $\eta(\boldsymbol{x}) = 1/2$. ∎

# B   Proof of Lemma 2

For $k \geq 2$, let $X_1, X_2, \cdots, X_{k-1}$ denote $k-1$ independent Bernoulli random variables with $X_i \sim \mathcal{B}(1/i)$ for $i \in [k-1]$. For any instance $\boldsymbol{x} \in \mathcal{X}$, we have

$$h(C(\boldsymbol{x})) = \sum_{i=1}^{k-1} X_i \quad \text{and} \quad E_{X_1, X_2, \cdots, X_k}[h(C(\boldsymbol{x}))] = \sum_{i=1}^{k-1} \frac{1}{i} \;.$$

Based on Lemma 6, we have

$$\ln k \leq E[h(C_i)] \leq 1 + \ln(k-1) \;.$$

It follows that, for any $\lambda < 0$ and by Markov's inequality,

$$\Pr\left[\sum_{i=1}^{k-1} X_i - E[X_i] \leq -\epsilon\right] \leq \exp\left(\lambda\epsilon - \lambda\sum_{i=1}^{k-1}\frac{1}{i}\right) E\left[\exp\left(\sum_{i=1}^{k-1}\lambda X_i\right)\right] \;. \tag{10}$$

We have, from the independence of random variables $X_1, X_2, \ldots, X_{k-1}$ with $X_i \sim \mathcal{B}(1/i)$,

$$E\left[\exp\left(\sum_{i=1}^{k-1}\lambda X_i\right)\right] = \prod_{i=1}^{k-1} E\left[\exp\left(\lambda X_i\right)\right] = \exp\left(\sum_{i=1}^{k-1}\ln\left(1 - \frac{1}{i} + \frac{1}{i}e^\lambda\right)\right) \;. \tag{11}$$

Denote by

$$g_i(\lambda) = \ln\left(1 - \frac{1}{i} + \frac{1}{i}e^\lambda\right) \;,$$

and we have

$$g_i'(\lambda) = \frac{e^\lambda}{i - 1 + e^\lambda} \;,$$

$$g_i''(\lambda) = \frac{e^\lambda}{i - 1 + e^\lambda} - \frac{e^{2\lambda}}{(i - 1 + e^\lambda)^2} \leq \frac{e^\lambda}{i - 1 + e^\lambda} < \frac{1}{i} \quad \text{for} \quad \lambda < 0.$$

Based on the Taylor expansions, we have

$$g_i(\lambda) \leq g_i(0) + \lambda g_i'(0) + \frac{\lambda^2}{2i} = \frac{\lambda}{i} + \frac{\lambda^2}{2i} \;.$$

Combining with Eqns. (10) and (11), this yields

$$\Pr\left[\sum_{i=1}^{k-1} X_i - \sum_{i=1}^{k-1}\frac{1}{i} \leq -\epsilon\right] \leq \exp\left(\lambda\epsilon + \frac{\lambda^2}{2}\sum_{i=1}^{k-1}\frac{1}{i}\right) \;.$$

By setting $\lambda = -\epsilon/\sum_{i=1}^{k-1} 1/i$, we have

$$\Pr\left[\sum_{i=1}^{k-1} X_i - \sum_{i=1}^{k-1}\frac{1}{i} \leq -\epsilon\right] \leq \exp\left(-\frac{\epsilon^2}{2\sum_{i=1}^{k-1} 1/i}\right) \;.$$

It follows that, by setting $\epsilon = \epsilon\sum_{i=1}^{k-1} 1/i$ and from Lemma 6,

$$\Pr_{X_1, X_2, \ldots, X_k}[h(C(\boldsymbol{x})) < (1-\epsilon)\ln k] \leq k^{-\epsilon^2/2} \;.$$

In a similar manner, we have, for any $\lambda > 0$,

$$\Pr\left[\sum_{i=1}^{k-1}(X_i - E[X_i]) \geq \epsilon\right]$$

$$\leq \exp\left(-\lambda\epsilon - \lambda\sum_{i=1}^{k-1}\frac{1}{i}\right) E\left[\exp\left(\sum_{i=1}^{k-1}\lambda X_i\right)\right]$$

$$\leq \exp\left(-\lambda\epsilon + \frac{\lambda^2}{2}\sum_{i=1}^{k-1}\frac{1}{i}\right) \;.$$

By setting $\lambda = \epsilon / \sum_{i=1}^{k-1} 1/i$, we have

$$\Pr\left[\sum_{i=1}^{k-1} X_i - \sum_{i=1}^{k-1} \frac{1}{i} \geq \epsilon\right] \leq \exp\left(-\frac{\epsilon^2}{2\sum_{i=1}^{k-1} 1/i}\right) .$$

It follows that, by setting $\epsilon = \epsilon \sum_{i=1}^{k-1} 1/i$ and combining with Lemma 6,

$$\Pr_{X_1, X_2, \dots, X_{k-1}} [h(C(\boldsymbol{x})) \geq (1 + \epsilon)(1 + \ln(k-1))] \leq k^{-\epsilon^2/2} ,$$

which completes the proof. ∎

## C  Proof of Lemma 3

Given any instance $\boldsymbol{x} \in \mathcal{X}$, recall that $C(\boldsymbol{x})$ denotes the rectangular cell containing instance $\boldsymbol{x}$, and $X_1, X_2, \cdots, X_{k-1}$ characterize the random events that the node containing instance $\boldsymbol{x}$ was selected for splitting in the construction of random tree, where $X_i \sim \mathcal{B}(1/i)$.

For $j \in [d]$, let $\ell_j(C(\boldsymbol{x}))$ denote the length of the $j$-th dimension of rectangular cell $C(\boldsymbol{x})$, and it is necessary to introduce the following random variables to analyze $\ell_j(C(\boldsymbol{x}))$.

- Let $Y_{1,j}, Y_{2,j}, \cdots, Y_{k-1,j}$ denote $k-1$ Bernoulli random variables such that $Y_{i,j} \sim \mathcal{B}(1/d)$ for $i \in [k-1]$. Here, $Y_{i,j} = 1$ denotes the random event that the $j$-th coordinate of the node, that contains the instance $\boldsymbol{x}$, is selected for random partition under the condition $X_i = 1$. We use $Y_{i,j}$ to illustrate the selection of coordinates with identical probability.
- Let $U_{1,j}, U_{2,j}, \cdots, U_{k-1,j}$ denote $k-1$ random variables with uniform distribution over $[0,1]$, i.e., $U_{i,j} \sim \mathcal{U}[0,1]$ for $i \in [k-1]$. Here, we use random variable $U_{i,j}$ to characterize the uniform and random splitting of the $j$-th coordinate of the node containing $\boldsymbol{x}$ under the condition $X_i Y_{i,j} = 1$ during the $i$-th construction of random tree.

It is easy to upper and lower bound $\ell_j(C(\boldsymbol{x}))$ as follows:

$$\prod_{i=1}^{k-1} \min(1 - U_{i,j}, U_{i,j})^{X_i Y_{i,j}} \leq \ell_j(C(\boldsymbol{x})) \leq \prod_{i=1}^{k-1} \max(1 - U_{i,j}, U_{i,j})^{X_i Y_{i,j}} . \tag{12}$$

**Lemma 8** *For $k \geq 2$ and $j \in [d]$, we have*

$$E\left[\prod_{i=1}^{k-1} (\max(U_{i,j}, 1 - U_{i,j}))^{X_i Y_{i,j}}\right] = \prod_{i=1}^{k-1} \left(1 - \frac{1}{4id}\right) \leq \exp\left(-\frac{\ln k}{4d}\right) , \tag{13}$$

$$E\left[\prod_{i=1}^{k-1} (\min(U_{i,j}, 1 - U_{i,j}))^{X_i Y_{i,j}}\right] = \prod_{i=1}^{k-1} \left(1 - \frac{3}{4id}\right) \geq \exp\left(-\frac{9 + 3\ln(k-1)}{4d}\right) , \tag{14}$$

*and we also have, for any instance $\boldsymbol{x} \in \mathcal{X}$,*

$$\exp\left(-\frac{9 + 3\ln(k-1)}{4d}\right) \leq E[\ell_j(C(\boldsymbol{x}))] \leq \exp\left(-\frac{\ln k}{4d}\right) .$$

*Here, all expectations take over independent random variables $X_1, \dots, X_{k-1}, Y_{1,j}, \dots, Y_{k-1,j}$ and $U_{1,j}, \dots, U_{k-1,j}$ with $X_i \sim \mathcal{B}(1/i)$, $Y_{i,j} \sim \mathcal{B}(1/d)$ and $U_{i,j} \sim \mathcal{U}(0,1)$ for $i \in [k-1]$.*

*Proof:*  For Eqn. (13), we first write $Z_{i,j} = (\max(U_{i,j}, 1 - U_{i,j}))^{X_i Y_{i,j}}$, and it follows that

$$E_{X_i, Y_{i,j}, U_{i,j}}[Z_{i,j}] = 1 - \frac{1}{id} + \frac{1}{id} E_{U_{i,j}}[\max(U_{i,j}, 1 - U_{i,j})] = 1 - \frac{1}{4id} ,$$

by using the fact

$$E_{U_{i,j}}[\max(U_{i,j}, 1 - U_{i,j})] = \int_0^1 \max(U_{i,j}, 1 - U_{i,j}) dU_{i,j}$$

$$= \int_0^{1/2} (1 - U_{i,j}) dU_{i,j} + \int_{1/2}^1 U_{i,j} dU_{i,j} = \frac{3}{4} .$$

It holds that, from Lemma 6 and by using the fact $1 - x \le e^{-x}$,

$$E\left[\prod_{i=1}^{k-1}(\max(U_{i,j}, 1 - U_{i,j}))^{X_i Y_{i,j}}\right] = \prod_{i=1}^{k-1}\left(1 - \frac{1}{4id}\right) \le \exp\left(-\frac{1}{4d}\sum_{i=1}^{k-1}\frac{1}{i}\right) \le \exp\left(-\frac{\ln k}{4d}\right) .$$

In a similar manner, we have

$$E_{X_i, Y_{i,j}, U_{i,j}}\left[(\min(U_{i,j}, 1 - U_{i,j}))^{X_i Y_{i,j}}\right] = 1 - \frac{1}{id} + \frac{1}{id}E_{U_{i,j}}[\min(U_{i,j}, 1 - U_{i,j})] = 1 - \frac{3}{4id} ,$$

by using the fact

$$
\begin{aligned}
E_{U_{i,j}}[\min(U_{i,j}, 1 - U_{i,j})] &= \int_0^1 \min(U_{i,j}, 1 - U_{i,j})dU_{i,j} \\
&= \int_0^{1/2} U_{i,j}dU_{i,j} + \int_{1/2}^1 (1 - U_{i,j})dU_{i,j} = \frac{1}{4} .
\end{aligned}
$$

It follows that

$$E\left[\prod_{i=1}^{k-1}(\min(U_{i,j}, 1 - U_{i,j}))^{X_i Y_{i,j}}\right] = \prod_{i=1}^{k-1}\left(1 - \frac{3}{4id}\right) = \exp\left(\sum_{i=1}^{k-1}\ln\left(1 - \frac{3}{4id}\right)\right) ,$$

which completes the proof of Eqn. (14) by combining with Lemma 7. ∎

**Lemma 9** *For integer $k \ge 2$, $j \in [d]$ and real $\epsilon > -1$, we have*

$$\Pr\left[\prod_{i=1}^{k-1}\left(\max(U_{i,j}, 1 - U_{i,j})\right)^{X_i Y_{i,j}} \ge (1 + \epsilon)\exp\left(-\frac{\ln k}{8d}\right)\right] \le \frac{e}{(1 + \epsilon)k^{1/8d}} ,$$

*where the probability takes over random variables $X_1, \ldots, X_{k-1}$, $Y_{1,j}, \ldots, Y_{k-1,j}$ and $U_{1,j}$, $\ldots, U_{k-1,j}$ with $X_i \sim \mathcal{B}(1/i)$, $Y_{i,j} \sim \mathcal{B}(1/d)$ and $U_{i,j} \sim \mathcal{U}(0,1)$ for $i \in [k-1]$.*

*Proof:* Based on the Markov's inequality and Lemma 8, we have, for any $\lambda > 0$,

$$
\begin{aligned}
&\Pr\left[\prod_{i=1}^{k-1}(\max(U_{i,j}, 1 - U_{i,j}))^{X_i Y_{i,j}} \ge (1 + \epsilon)\exp\left(-\frac{\ln k}{8d}\right)\right] \\
&= \Pr\left[\prod_{i=1}^{k-1}(\max(U_{i,j}, 1 - U_{i,j}))^{\lambda X_i Y_{i,j}} \ge (1 + \epsilon)^\lambda\left(\exp\left(-\frac{\ln k}{8d}\right)\right)^\lambda\right] \\
&\le (1 + \epsilon)^{-\lambda}\exp\left(\frac{\lambda \ln k}{8d}\right) \times E\left[\prod_{i=1}^{k-1}(\max(U_{i,j}, 1 - U_{i,j}))^{\lambda X_i Y_{i,j}}\right] .
\end{aligned}
$$

Let $Z_{i,j} = (\max(U_{i,j}, 1 - U_{i,j}))^{\lambda X_i Y_{i,j}}$, and we have

$$
\begin{aligned}
&E_{X_i \sim \mathcal{B}(1/i), Y_{i,j} \sim \mathcal{B}(1/d), U_{i,j} \sim \mathcal{U}(0,1)}[Z_{i,j}] \\
&= 1 - \frac{1}{id} + \frac{1}{id}E_{U_{i,j} \sim \mathcal{U}(0,1)}[(\max(U_{i,j}, 1 - U_{i,j}))^\lambda] \\
&\le 1 - \frac{1}{id} + \frac{2 - 1/2^\lambda}{id(\lambda + 1)} ,
\end{aligned}
$$

where the last equation holds from

$$E_{U_{i,j} \sim \mathcal{U}(0,1)}[(\max(U_{i,j}, 1 - U_{i,j}))^\lambda] = \int_0^{1/2}(1 - U_{i,j})^\lambda dU_{i,j} + \int_{1/2}^1 U_{i,j}^\lambda dU_{i,j} = \frac{2 - 1/2^\lambda}{\lambda + 1} .$$

It follows that, by using $1 + x \le e^x$,

$$E\left[\prod_{i=1}^{k-1}(\max(U_{i,j}, 1 - U_{i,j}))^{\lambda X_i Y_{i,j}}\right] \le \exp\left(-\sum_{i=1}^{k-1}\frac{1}{id} + \sum_{i=1}^{k-1}\frac{2 - 1/2^\lambda}{(\lambda + 1)id}\right) .$$

Based on Lemma 6, we have

$$E\left[\prod_{i=1}^{k-1}(\max(U_{i,j}, 1-U_{i,j}))^{\lambda X_i Y_{i,j}}\right] \leq \exp\left(-\frac{\ln k}{d} + \frac{(2-1/2^\lambda)(1+\ln(k-1))}{(\lambda+1)d}\right) .$$

In a summary, we have

$$\Pr\left[\prod_{i=1}^{k-1}(\max(U_{i,j}, 1-U_{i,j}))^{X_i Y_{i,j}} \geq (1+\epsilon)\exp\left(\frac{\ln k}{8d}\right)\right]$$

$$\leq \exp\left(-\lambda\ln(1+\epsilon) - \frac{\ln k}{d} + \frac{\lambda \ln k}{8d} + \frac{(2-1/2^\lambda)(1+\ln(k-1))}{(\lambda+1)d}\right) .$$

By setting $\lambda = 1$, we have

$$\Pr\left[\prod_{i=1}^{k-1}(\max(U_{i,j}, 1-U_{i,j}))^{X_i Y_{i,j}} \geq (1+\epsilon)\exp\left(\frac{\ln k}{8d}\right)\right]$$

$$\leq \exp\left(-\ln(1+\epsilon) - \frac{7\ln k}{8d} + \frac{3(1+\ln(k-1))}{4d}\right)$$

$$\leq \frac{e^{3/4d}}{(1+\epsilon)^{1/8d}} ,$$

which completes the proof for dimension $d \geq 1$. ∎

**Proof of Lemma 3.** Based on the union bounds, we have

$$\Pr\left[\nu[C(\boldsymbol{x})] \geq \frac{(1+\epsilon)\sqrt{d}}{k^{1/8d}}\right]$$

$$= \Pr\left[\nu[C(\boldsymbol{x})] \geq (1+\epsilon)\sqrt{d}\exp\left(-\frac{\ln k}{8d}\right)\right]$$

$$\leq \Pr\left[\exists j \in [d] : \ell_j(C(\boldsymbol{x})) \geq (1+\epsilon)\exp\left(-\frac{\ln k}{8d}\right)\right]$$

$$\leq d\Pr\left[\ell_1(C(\boldsymbol{x})) \geq (1+\epsilon)\exp\left(-\frac{\ln k}{8d}\right)\right]$$

$$\leq d\Pr\left[\prod_{i=1}^{k-1}\left(\max(U_{i,1}, 1-U_{i,1})\right)^{X_i Y_{i,1}} \geq (1+\epsilon)\exp\left(-\frac{\ln k}{8d}\right)\right]$$

$$\leq \frac{ed}{(1+\epsilon)k^{1/8d}} ,$$

where the last inequality holds from Lemma 9. This completes the proof. ∎

## D  Proof of Lemma 4

It is necessary to introduce two lemmas as follows:

**Lemma 10** *For any rectangular cell $C_i \subseteq \mathcal{X}$, we have*

$$\Pr[\boldsymbol{x} \in C_i]\Pr[|C_i \cap S_n| < n\Pr[\boldsymbol{x} \in C_i]/2] \leq 3/n .$$

*Proof:* From Lemma 5, we have

$$\Pr[|C_i \cap S_n| < n\Pr[\boldsymbol{x} \in C_i]/2] \leq \exp(-n\Pr[\boldsymbol{x} \in C_i]/8) ,$$

and it holds that

$$\Pr[\boldsymbol{x} \in C_i]\Pr[|C_i \cap S_n| < n\Pr[\boldsymbol{x} \in C_i]/2] \leq \frac{8}{ne} \leq \frac{3}{n}$$

by using $\max_x xe^{-ax} \leq 1/ae$. This completes the proof. ∎

**Lemma 11** *Let $X_1, X_2, \ldots, X_m$ be $m$ independent random variables with $X_i \sim \mathcal{B}(\eta_i)$ for $i \in [m]$, and set $\kappa = \sum_{i=1}^{m} \eta_i / m$. For $\kappa \in [0, 1/2)$, we have*

$$(1 - 2\kappa) \Pr_{X_1 \sim \mathcal{B}(\eta_1), \ldots, X_m \sim \mathcal{B}(\eta_m)} \left[ \sum_{i=1}^{m} X_i \geq \frac{m}{2} \right] \leq \frac{1}{\sqrt{2m}} . \tag{15}$$

*For $\kappa \in [1/2, 1]$, we also have*

$$(2\kappa - 1) \Pr_{X_1 \sim \mathcal{B}(\eta_1), \ldots, X_m \sim \mathcal{B}(\eta_m)} \left[ \sum_{i=1}^{m} X_i < \frac{m}{2} \right] \leq \frac{1}{\sqrt{2m}} . \tag{16}$$

*Proof:* For any $\lambda > 0$, we have, from the Markov's inequality,

$$\Pr_{X_1 \sim \mathcal{B}(\eta_1), \ldots, X_m \sim \mathcal{B}(\eta_m)} \left[ \sum_{i=1}^{m} X_i \geq \frac{m}{2} \right]$$

$$\leq \exp(-m\lambda/2) \mathop{E}_{X_1 \sim \mathcal{B}(\eta_1), \ldots, X_m \sim \mathcal{B}(\eta_m)} \left[ \exp\left( \lambda \sum_{i=1}^{m} X_i \right) \right]$$

$$= \exp(-m\lambda/2) \prod_{i=1}^{m} \mathop{E}_{X_i \sim \mathcal{B}(\eta_i)} [\exp(\lambda X_i)] .$$

From $X_i \sim \mathcal{B}(\eta_i)$, we have

$$E[\exp(\lambda X_i)] = 1 - \eta_i e^0 + \eta_i e^\lambda \leq \exp(\eta_i(e^\lambda - 1)) .$$

Write $\kappa = \sum_{i=1}^{m} \eta_i / m$, and it holds that

$$\Pr_{X_1 \sim \mathcal{B}(\eta_1), \ldots, X_m \sim \mathcal{B}(\eta_m)} \left[ \sum_{i=1}^{m} X_i \geq \frac{m}{2} \right] \leq \exp(-m\lambda/2 + m\kappa(e^\lambda - 1)) .$$

By setting $\lambda = -\ln(2\kappa)$, we have

$$\Pr_{X_1 \sim \mathcal{B}(\eta_1), \ldots, X_m \sim \mathcal{B}(\eta_m)} \left[ \sum_{i=1}^{m} X_i \geq \frac{m}{2} \right] \leq \exp(m/2 + m\ln(2\kappa)/2 - m\kappa) . \tag{17}$$

We introduce another function

$$g_1(\kappa) = (1 - 2\kappa) \exp(m/2 + m\ln(2\kappa)/2 - m\kappa) , \tag{18}$$

and the derivative is given by

$$g_1'(\kappa) = \exp(m/2 + m\ln(2\kappa)/2 - m\kappa)(2\kappa m - 2m - 2 + m/2\kappa) .$$

Solving $g_1'(\kappa) = 0$ gives the optimal solution

$$\kappa^* = \frac{1}{2} - \frac{1}{1 + \sqrt{2m + 1}} .$$

It is easy to find that, for continuous function $g(\kappa)$ with $\kappa \in [0, 1/2)$

$$g_1(\kappa) \leq \max_{\kappa \in [0, 1/2)} g_1(\kappa) = \max\{g_1(0), g_1(1/2), g_1(\kappa^*)\} = g_1(\kappa^*) , \tag{19}$$

and we further have

$$g_1(\kappa^*) = \frac{1}{1 + \sqrt{1 + 2m}} \exp\left( \frac{m}{1 + \sqrt{1 + 2m}} + \frac{m}{2} \ln\left( 1 - \frac{2}{1 + \sqrt{1 + 2m}} \right) \right)$$

$$\leq \frac{1}{1 + \sqrt{1 + 2m}} \leq \frac{1}{\sqrt{2m}} ,$$

where the first inequality holds from $\ln(1 - x) \leq -x$. Hence, Eqn. (15) holds from Eqns. (17)-(19).

For Eqn. (16), we similarly have, by using Markov's inequality,

$$\Pr_{X_1\sim\mathcal{B}(\eta_1),\ldots,X_m\sim\mathcal{B}(\eta_m)}\left[\sum_{i=1}^m X_i < \frac{m}{2}\right] \leq \exp(-m\lambda/2 + m\kappa(e^\lambda - 1))$$

for $\lambda \leq 0$. By setting $\lambda = -\ln(2\kappa)$ for $\kappa \in [1/2, 1]$, we have

$$\Pr_{X_1\sim\mathcal{B}(\eta_1),\ldots,X_m\sim\mathcal{B}(\eta_m)}\left[\sum_{i=1}^m X_i < \frac{m}{2}\right] \leq \exp(m/2 + m\ln(2\kappa)/2 - m\kappa). \tag{20}$$

We also introduce another function

$$g_2(\kappa) = (2\kappa - 1)\exp(m/2 + m\ln(2\kappa)/2 - m\kappa), \tag{21}$$

and solving $g_2'(\kappa) = 0$ gives the optimal solution

$$\kappa^* = \frac{1}{2} + \frac{1}{1 + \sqrt{2m+1}}.$$

It is easy to find that, for continuous function $g(\kappa)$ with $\kappa \in [1/2, 1]$,

$$g_2(\kappa) \leq \max_{\kappa\in[0,1/2)} g(\kappa) = \max\{g_2(1/2), g_2(1), g(\kappa^*)\} = g_2(\kappa^*) \leq 1/\sqrt{2m}.$$

This proves Eqn. (16) by combining with Eqns. (20) and (21). ∎

**Proof of Lemma 4.** This lemma holds obviously when $\Pr[\boldsymbol{x} \in C_i] = 0$, and it suffices to consider $\Pr[\boldsymbol{x} \in C_i] > 0$. We introduce the random events

$$\begin{aligned}\Gamma_1 &= \{|C_i \cap S_n| \geq n\Pr[\boldsymbol{x}\in C_i]/2\}, \\ \Gamma_2 &= \{|C_i \cap S_n| < n\Pr[\boldsymbol{x}\in C_i]/2\}.\end{aligned}$$

Based on the law of total probability, we have

$$\begin{aligned}&\Pr_{S_n,(\boldsymbol{x},y)}[f_{\Theta,S_n}(\boldsymbol{x}) \neq y | \boldsymbol{x}\in C_i]\\ &= \Pr_{S_n,(\boldsymbol{x},y)}[f_{\Theta,S_n}(\boldsymbol{x}) \neq y | \boldsymbol{x}\in C_i, \Gamma_1]\Pr[\Gamma_1] + \Pr_{S_n,(\boldsymbol{x},y)}[f_{\Theta,S_n}(\boldsymbol{x}) \neq y | \boldsymbol{x}\in C_i, \Gamma_2]\Pr[\Gamma_2].\end{aligned}$$

It follows that, from Lemma 10,

$$\begin{aligned}&\Pr_{S_n,(\boldsymbol{x},y)}[f_{\Theta,S_n}(\boldsymbol{x}) \neq y | \boldsymbol{x}\in C_i]\Pr[\boldsymbol{x}\in C_i]\\ &\leq \Pr_{S_n,(\boldsymbol{x},y)}[f_{\Theta,S_n}(\boldsymbol{x}) \neq y | \boldsymbol{x}\in C_i, \Gamma_1]\Pr[\boldsymbol{x}\in C_i]\Pr[\Gamma_1] + 3/n. \tag{22}\end{aligned}$$

To bound the term $\Pr_{S_n,(\boldsymbol{x},y)}[f_{\Theta,S_n}(\boldsymbol{x}) \neq y | \boldsymbol{x}\in C_i, \Gamma_1]$, we further introduce the set $S_n^i$ of training examples as follows:

$$S_n^i = \{(\boldsymbol{x}_j, y_j)\colon (\boldsymbol{x}_j, y_j) \in S_n \text{ and } \boldsymbol{x}_j \in C_i\},$$

i.e., the training examples falling into the cell $C_i$. Under the condition $\Gamma_1$, we have

$$m := |S_n^i| = |S_n \cap C_i| \geq n\Pr[C_i]/2. \tag{23}$$

Without loss of generality, we denote by $S_n^i = \{(\boldsymbol{x}_1, y_1), (\boldsymbol{x}_2, y_2), \ldots, (\boldsymbol{x}_m, y_m)\}$. For instance $\boldsymbol{x} \in C_i$, its label can be predicted by random forests classifier as

$$f_{\Theta,S_n}(\boldsymbol{x}) = I\left[\sum_{j=1}^m y_j \geq m/2\right].$$

Conditioned on $\boldsymbol{x}, \boldsymbol{x}_1, \boldsymbol{x}_2, \ldots, \boldsymbol{x}_m$, we can observe that $y \sim \mathcal{B}(\eta(\boldsymbol{x}))$ and $y_j \sim \mathcal{B}(\eta(\boldsymbol{x}_j))$ for $j \in [m]$, and set $\kappa = \sum_{j=1}^m \eta(\boldsymbol{x}_j)/m$. It follows that

$$\begin{aligned}&\Pr_{y_1,\ldots,y_m,y}[f_{\Theta,S_n}(\boldsymbol{x}) \neq y | \boldsymbol{x}_1, \ldots, \boldsymbol{x}_n]\\ &= \eta(\boldsymbol{x})\Pr_{y_1,\ldots,y_m}\left[\sum_{j=1}^m y_j < \frac{m}{2}\right] + (1 - \eta(\boldsymbol{x}))\Pr_{y_1,\ldots,y_m}\left[\sum_{j=1}^m y_j \geq \frac{m}{2}\right]. \tag{24}\end{aligned}$$

If $\kappa \in [0, 1/2)$, then we have, from Eqn. (24)

$$\Pr_{y_1,\dots,y_m,y} [f_{\Theta,S_n}(\boldsymbol{x}) \neq y] = \eta(\boldsymbol{x}) + (1 - 2\eta(\boldsymbol{x})) \Pr_{y_1,\dots,y_m} \left[ \sum_{j=1}^{m} y_j \geq \frac{m}{2} \right] ,$$

and it follows that:

- If $\eta(x) = 1/2$, then we have $1 - 2\eta(x) = 0$ and
  $$\Pr_{y_1,\dots,y_m,y} [f_{\Theta,S_n}(\boldsymbol{x}) \neq y] = \eta(\boldsymbol{x}) = \min\{\eta(\boldsymbol{x}), 1 - \eta(\boldsymbol{x})\} .$$

- If $\eta(x) > 1/2$, then we have $1 - 2\eta(x) < 0$, and for $\kappa \in [0, 1/2)$, we also have
  $$\Pr_{y_1,\dots,y_m,y} [f_{\Theta,S_n}(\boldsymbol{x}) \neq y] < \eta(\boldsymbol{x}) \quad = \quad \min\{\eta(\boldsymbol{x}), 1 - \eta(\boldsymbol{x})\} + 2\eta(\boldsymbol{x}) - 1$$
  $$\leq \quad \min\{\eta(\boldsymbol{x}), 1 - \eta(\boldsymbol{x})\} + 2|\eta(\boldsymbol{x}) - \kappa| .$$

- If $\eta(x) < 1/2$, then we have
  $$\Pr_{y_1,\dots,y_m,y} [f_{\Theta,S_n}(\boldsymbol{x}) \neq y]$$
  $$\leq \quad \min\{\eta(\boldsymbol{x}), 1 - \eta(\boldsymbol{x})\} + 2|\eta(\boldsymbol{x}) - \kappa| + (1 - 2\kappa) \Pr_{y_1,\dots,y_m} \left[ \sum_{j=1}^{m} y_j \geq \frac{m}{2} \right]$$
  $$\leq \quad \min\{\eta(\boldsymbol{x}), 1 - \eta(\boldsymbol{x})\} + 2|\eta(\boldsymbol{x}) - \kappa| + 1/\sqrt{2m} ,$$
  where the last inequality holds from $\kappa \in [0, 1/2)$ and Eqn. (15) in Lemma 11.

In a summary, we have, for $\kappa \in [0, 1/2)$,

$$\Pr_{y_1,\dots,y_m,y} [f_{\Theta,S_n}(\boldsymbol{x}) \neq y] \leq \min\{\eta(\boldsymbol{x}), 1 - \eta(\boldsymbol{x})\} + 2|\eta(\boldsymbol{x}) - \kappa| + 1/\sqrt{2m} .$$

In a similar manner, we have, for $\kappa \in [1/2, 1]$,

$$\Pr_{y_1,\dots,y_m,y} [f_{\Theta,S_n}(\boldsymbol{x}) \neq y] \quad = \quad 1 - \eta(\boldsymbol{x}) + (2\eta(\boldsymbol{x}) - 1) \Pr_{y_1,\dots,y_m} \left[ \sum_{j=1}^{m} y_j \geq \frac{m}{2} \right]$$
$$\leq \quad \min\{\eta(\boldsymbol{x}), 1 - \eta(\boldsymbol{x})\} + 2|\eta(\boldsymbol{x}) - \kappa| + 1/\sqrt{2m} .$$

From the $L$-Lipschtiz assumption, we have, for $\boldsymbol{x}, \boldsymbol{x}_1, \dots, \boldsymbol{x}_m \in C_i$,

$$|\eta(\boldsymbol{x}) - \kappa| = \left| \eta(\boldsymbol{x}) - \sum_{j=1}^{m} \frac{\eta(\boldsymbol{x}_j)}{m} \right| \leq \sum_{j=1}^{m} |\eta(\boldsymbol{x}) - \eta(\boldsymbol{x}_j)| / m \leq L\nu(C_i) .$$

It follows that

$$\Pr_{y_1,\dots,y_m,y} [f_{\Theta,S_n}(\boldsymbol{x}) \neq y] \leq \min\{\eta(\boldsymbol{x}), 1 - \eta(\boldsymbol{x})\} + 2L\nu(C_i) + 1/\sqrt{2m} .$$

Hence, we have, from Eqn. (23)

$$\Pr_{S_n,(\boldsymbol{x},y)} [f_{\Theta,S_n}(\boldsymbol{x}) \neq y | \boldsymbol{x} \in C_i, \Gamma_1] \Pr[C_i] \Pr[\Gamma_1]$$
$$\leq \quad E_{\boldsymbol{x}}[\min\{\eta(\boldsymbol{x}), 1 - \eta(\boldsymbol{x})\} | \boldsymbol{x} \in C_i] \Pr[C_i] + 2L\nu(C_i) \Pr[C_i] + \sqrt{\Pr[C_i]/n} ,$$

which completes the proof by combining with Eqn. (22). ∎

## E    Proof of Theorem 2

We first introduce some lemmas before the proof of Theorem 2.

**Lemma 12** *For integer $k \geq 2$, $d \geq 2$, $j \in [d]$ and real $\epsilon > -1$, we have*

$$\Pr\left[\prod_{i=1}^{k-1}\left(\frac{1}{2}\right)^{X_i Y_{i,j}} \geq (1+\epsilon)\exp\left(-\frac{\ln k}{4d}\right)\right] \leq \frac{3/2}{(1+\epsilon)^{3/2}k^{1/3.6846d}} ,$$

*where the probability takes over random variables $X_1, \ldots, X_{k-1}, Y_{1,j}, \ldots, Y_{k-1,j}$ with $X_i \sim \mathcal{B}(1/i)$ and $Y_{i,j} \sim \mathcal{B}(1/d)$ for $i \in [k-1]$.*

*Proof:* For any $\lambda > 0$, we have, based on the Markov's inequality,

$$\Pr\left[\prod_{i=1}^{k-1}\left(\frac{1}{2}\right)^{X_i Y_{i,j}} \geq (1+\epsilon)\exp\left(-\frac{\ln k}{4d}\right)\right]$$

$$= \Pr\left[\prod_{i=1}^{k-1}\left(\frac{1}{2}\right)^{\lambda X_i Y_{i,j}} \geq (1+\epsilon)\exp\left(-\frac{\ln k}{4d}\right)\right]$$

$$\leq (1+\epsilon)^{-\lambda}\exp\left(\frac{\lambda \ln k}{4d}\right) \times E\left[\prod_{i=1}^{k-1}\left(\frac{1}{2}\right)^{\lambda X_i Y_{i,j}}\right] .$$

From $X_i \sim \mathcal{B}(1/i)$ and $Y_{i,j} \sim \mathcal{B}(1/d)$ ($i \in [k-1]$), we have, by using $1 + x \leq e^x$,

$$E\left[\prod_{i=1}^{k-1}\left(\frac{1}{2}\right)^{\lambda X_i Y_{i,j}}\right] = \prod_{i=1}^{k-1}\left(1 - \frac{1}{id} + \frac{1}{id2^\lambda}\right) \leq \exp\left(-\sum_{i=1}^{k-1}\frac{1}{id} + \sum_{i=1}^{k-1}\frac{1}{id2^\lambda}\right) ,$$

which yields that

$$\Pr\left[\prod_{i=1}^{k-1}\left(\frac{1}{2}\right)^{X_i Y_{i,j}} \geq (1+\epsilon)\exp\left(-\frac{\ln k}{4d}\right)\right]$$

$$\leq \exp\left(-\lambda\ln(1+\epsilon) + \frac{\lambda \ln k}{4d} - \sum_{i=1}^{k-1}\frac{1}{id} + \sum_{i=1}^{k-1}\frac{1}{id2^\lambda}\right) .$$

By setting $\lambda = 3/2$, we have

$$\Pr\left[\prod_{i=1}^{k-1}(\max(U_i, 1-U_i))^{X_i Y_{i,j}} \geq (1+\epsilon)\exp\left(-\frac{\ln k}{4d}\right)\right]$$

$$\leq \exp\left(-\frac{3}{2}\ln(1+\epsilon) - \frac{5\ln k}{8d} + \frac{1+\ln(k-1)}{2\sqrt{2}d}\right)$$

$$\leq \frac{e^{1/(2\sqrt{2}d)}}{(1+\epsilon)^{3/2}k^{1/3.6846d}} ,$$

which completes the proof by using $e^{1/(2\sqrt{2}d)} \leq 3/2$. ∎

Based on Lemma 12, we can bound the diameter $\nu(C(\boldsymbol{x}))$ as follows:

**Lemma 13** *For real $\epsilon > -1$ and instance $\boldsymbol{x} \in \mathcal{X}$, we have*

$$\Pr\left[\nu[C(\boldsymbol{x})] \geq (1+\epsilon)\sqrt{d}/k^{1/4d}\right] \leq \frac{3d/2}{(1+\epsilon)^{3/2}k^{1/3.6846d}} ,$$

*where the probability takes over random selection of splitting leaves and dimensions.*

*Proof:* For $j \in [d]$, recall that $\ell_j(C(\boldsymbol{x}))$ denotes the length of the $j$-th coordinate of $C(\boldsymbol{x})$ for $j \in [d]$. Let $X_1, \ldots, X_{k-1}, Y_{1,j}, \ldots, Y_{k-1,j}$ be random variables with $X_i \sim \mathcal{B}(1/i)$ and $Y_{i,j} \sim \mathcal{B}(1/d)$ for $i \in [k-1]$. According to the construction of pure random tree with midpoint split, we have

$$\ell_j(C(\boldsymbol{x})) = 1/2^{X_i Y_{i,j}} .$$

Based on Lemma 12, we have

$$\Pr\left[\ell_j(C(\boldsymbol{x})) \geq (1+\epsilon)\exp\left(-\frac{\ln k}{4d}\right)\right] \leq \frac{3/2}{(1+\epsilon)^{3/2}k^{1/3.6846d}} . \tag{25}$$

Based on union bounds, we have

$$\Pr\left[\nu[C(\boldsymbol{x})] \geq (1+\epsilon)\sqrt{d}/k^{1/4d}\right]$$

$$= \Pr\left[\nu[C(\boldsymbol{x})] \geq (1+\epsilon)\sqrt{d}\exp\left(-\frac{\ln k}{4d}\right)\right]$$

$$\leq \Pr\left[\exists j \in [d] : \ell_j(C(\boldsymbol{x})) \geq (1+\epsilon)\exp\left(-\frac{\ln k}{4d}\right)\right]$$

$$\leq d\Pr\left[\ell_1(C(\boldsymbol{x})) \geq (1+\epsilon)\exp\left(-\frac{\ln k}{4d}\right)\right]$$

$$\leq \frac{3d/2}{(1+\epsilon)^{3/2}k^{1/3.6846d}} ,$$

where the last inequality holds from Eqn. (25). ∎

**Proof of Theorem 2** Similarly to Theorem 1, we first study the convergence rate of individual random tree classifier $f_{S_n,\Theta}(\boldsymbol{x})$. Based on the law of total probability, we have

$$R_{\mathcal{D}}(f_{S_n,\Theta}) = \Pr_{(\boldsymbol{x},y)\sim\mathcal{D}}[f_{\Theta,S_n}(\boldsymbol{x}) \neq y] = E_{\boldsymbol{x}\sim\mathcal{D}_{\mathcal{X}}}\left[\Pr_{y\sim\mathcal{B}(\eta(\boldsymbol{x}))}[f_{\Theta,S_n}(\boldsymbol{x}) \neq y]\right] .$$

For random forests classifier $f_{\Theta,S_n}(\boldsymbol{x})$, we associate a set as follows

$$\Lambda_2 = \left\{\boldsymbol{x} \in \mathcal{X} : \nu(C(\boldsymbol{x})) \geq (1+\epsilon)\sqrt{d}/k^{1/4d}\right\} , \tag{26}$$

and it follows that

$$R_{\mathcal{D}}(f_{S_n,\Theta}) \leq E_{\boldsymbol{x}\sim\mathcal{D}_{\mathcal{X}}}\left[\mathbb{I}[\boldsymbol{x} \in \Lambda_2]\right] + E_{\boldsymbol{x}\sim\mathcal{D}_{\mathcal{X}}}\left[\Pr_{y\sim\mathcal{B}(\eta(\boldsymbol{x}))}[f_{\Theta,S_n}(\boldsymbol{x}) \neq y|\boldsymbol{x}]\mathbb{I}[\boldsymbol{x} \notin \Lambda_2]\right] . \tag{27}$$

Notice that $C_1, C_2, \ldots, C_k$ is a partition of the instance space $\mathcal{X}$, and we have

$$E_{\boldsymbol{x}\sim\mathcal{D}_{\mathcal{X}}}\left[\Pr_{y\sim\mathcal{B}(\eta(\boldsymbol{x}))}[f_{\Theta,S_n}(\boldsymbol{x}) \neq y]\mathbb{I}[\boldsymbol{x} \notin \Lambda_2]\right]$$

$$= \sum_{i=1}^{k}\Pr[f_{\Theta,S_n}(\boldsymbol{x}) \neq y|\boldsymbol{x} \in C_i]\Pr[\boldsymbol{x} \in C_i]\mathbb{I}[C_i \nsubseteq \Lambda_2] ,$$

where we use the fact $C(\boldsymbol{x}) = C_i$ for every $\boldsymbol{x} \in C_i$. It follows that, from Eqns. (26) and (27),

$$E_{S_n,\Theta}\left[R_{\mathcal{D}}(f_{S_n,\Theta})\right]$$

$$\leq E_{\boldsymbol{x}\sim\mathcal{D}_{\mathcal{X}}}\left[\Pr_{S_n,\Theta}\left[\nu[C(\boldsymbol{x})] \geq \frac{(1+\epsilon)\sqrt{d}}{(1+k)^{1/4d}}\right]\right] \tag{28}$$

$$+ E_{\Theta}\left[\sum_{i=1}^{k}E_{S_n}[\Pr[f_{\Theta,S_n}(\boldsymbol{x}) \neq y|\boldsymbol{x} \in C_i]]\Pr[\boldsymbol{x} \in C_i]\mathbb{I}[C_i \nsubseteq \Lambda_2]\right] . \tag{29}$$

From Lemma 13, Eqn. (28) can be further upper bounded by

$$E_{\boldsymbol{x}\sim\mathcal{D}_{\mathcal{X}}}\left[\Pr_{S_n,\Theta}\left[\nu[C(\boldsymbol{x})] \geq (1+\epsilon)\sqrt{d}/k^{1/4d}\right]\right] \leq \frac{3d/2}{(1+\epsilon)^{3/2}k^{1/3.6846d}} . \tag{30}$$

Based on Lemma 4 and Eqn. (26), we can bound the term in Eqn. (29) as

$$\sum_{i=1}^{k}E_{S_n}[\Pr[f_{\Theta,S_n}(\boldsymbol{x}) \neq y|\boldsymbol{x} \in C_i]]\Pr[\boldsymbol{x} \in C_i]\mathbb{I}[C_i \nsubseteq \Lambda_2]$$

$$\leq R_{\mathcal{D}}^* + \frac{2(1+\epsilon)L\sqrt{d}}{k^{1/4d}} + \sqrt{\frac{k}{n}} + \frac{3k}{n} . \tag{31}$$

It follows that, by combining with Eqns. (28)-(31),

$$E_{S_n,\Theta}\left[R_{\mathcal{D}}(f_{S_n,\Theta})\right] \le R_{\mathcal{D}}^* + \frac{3d/2}{(1+\epsilon)^{3/2}k^{1/3.6846d}} + \frac{2(1+\epsilon)L\sqrt{d}}{k^{1/4d}} + \sqrt{\frac{k}{n}} + \frac{3k}{n} \ .$$

By setting

$$\epsilon = \left(\frac{9\sqrt{d}}{8L}k^{\frac{1}{4d}-\frac{1}{3.6848d}}\right)^{2/5} - 1 \ ,$$

we have, by simple algebraic calculations,

$$E_{S_n,\Theta}\left[R_{\mathcal{D}}(f_{S_n,\Theta})\right] \le R_{\mathcal{D}}^* + \frac{4L^{3/5}d^{7/10}}{k^{1/3.87d}} + \sqrt{\frac{k}{n}} + \frac{3k}{n} \ ,$$

which completes the proof by combining with Lemma 1. ∎

## F   Proof of Theorem 3

We begin with a lemma as follows:

**Lemma 14** *Let $S_n$ be a training data drawn i.i.d. from distribution $\mathcal{D}$. For any rectangle cell $C \subseteq \mathcal{X}$ and integer $\tau \ge 2$, we have*

$$\Pr[\boldsymbol{x} \in C]\Pr[|C \cap S_n| \le \tau] \le \frac{\tau}{n}\left(1 + \sqrt{\frac{2}{\tau}}\right) \ .$$

*Proof:*  For any $\delta \in (0,1)$, if $\tau \le (1-\delta)n\Pr[\boldsymbol{x} \in C]$, then we have, based on Lemma 5,

$$\Pr[|C \cap S_n| \le \tau] \ \le \ \Pr[|C \cap S_n| \le (1-\delta)n\Pr[\boldsymbol{x} \in C]] \ \le \ \exp(-n\Pr[\boldsymbol{x} \in C]\delta^2/2) \ .$$

It follows that, by using $\max_x xe^{-ax} = 1/ae$,

$$\Pr[\boldsymbol{x} \in C]\Pr[|C \cap S_n| \le \tau] \le \Pr[\boldsymbol{x} \in C]\exp(-n\Pr[\boldsymbol{x} \in C]\delta^2/2) \le \frac{2}{ne\delta^2} \ .$$

If $\tau \ge (1-\delta)n\Pr[\boldsymbol{x} \in C]$, then we have

$$\Pr[\boldsymbol{x} \in C]\Pr[|C \cap S_n| \le \tau] \le \Pr[\boldsymbol{x} \in C] \le \frac{\tau}{n(1-\delta)} \ .$$

By setting $\delta = (\sqrt{1+2\tau e} - 1)/\tau e$, we have

$$\frac{\tau}{n(1-\delta)} = \frac{2}{ne\delta^2} = \frac{\tau}{n} \times \frac{\tau e + 1 + \sqrt{2\tau e + 1}}{\tau e} \le \frac{\tau}{n}\left(1 + \sqrt{\frac{2}{\tau}}\right) \text{ for } \tau \ge 2,$$

which completes the proof. ∎

**Proof of Theorem 3**. Similarly to the proof of Theorem 1, we first present the convergence rate of individual random tree classifier $f_{S_n,\Theta}(\boldsymbol{x})$ according to Algorithm 1. Let $C_1, C_2, \ldots, C_k$ be a partition of instance space $\mathcal{X}$, which are associated with $k$ leaves of random tree. Based on the law of total probability, we have the classification error of random forests classifier $f_{\Theta,S_n}(\boldsymbol{x})$ with respect to distribution $\mathcal{D}$

$$\begin{aligned}
R_{\mathcal{D}}(f_{S_n,\Theta}) &= \Pr_{(\boldsymbol{x},y)\sim\mathcal{D}}[f_{\Theta,S_n}(\boldsymbol{x}) \ne y] \\
&= \sum_{i=1}^{k}\Pr[f_{\Theta,S_n}(\boldsymbol{x}) \ne y | \boldsymbol{x} \in C_i]\Pr[\boldsymbol{x} \in C_i] \ .
\end{aligned}$$

We introduce a set

$$\Lambda_3 = \{C_i : \text{ all training examples in } C_i \text{ have the same label}\}.$$

It follows that

$$R_{\mathcal{D}}(f_{S_n,\Theta}) \ = \ \sum_{i=1}^{k} \Pr[f_{\Theta,S_n}(\boldsymbol{x}) \neq y | \boldsymbol{x} \in C_i] \Pr[\boldsymbol{x} \in C_i] \mathbb{I}[C_i \in \Lambda_3] \tag{32}$$

$$+ \sum_{i=1}^{k} \Pr[f_{\Theta,S_n}(\boldsymbol{x}) \neq y | \boldsymbol{x} \in C_i] \Pr[\boldsymbol{x} \in C_i] \mathbb{I}[C_i \notin \Lambda_3] \ . \tag{33}$$

If $C_i \in \Lambda_3$, then we have, for $\tau \geq 2$,

$$\Pr[C_i] \Pr\left[f_{\Theta,S_n}(\boldsymbol{x}) \neq y \Big| \boldsymbol{x} \in C_i\right]$$
$$= \ \Pr[C_i] \Pr[f_{\Theta,S_n}(\boldsymbol{x}) \neq y, |C_i \cap S_n| \leq \tau | \boldsymbol{x} \in C_i]$$
$$+ \Pr[C_i] \Pr[f_{\Theta,S_n}(\boldsymbol{x}) \neq y, |C_i \cap S_n| > \tau | \boldsymbol{x} \in C_i]$$
$$\leq \ \Pr[C_i] \Pr[|C_i \cap S_n| \leq \tau]$$
$$+ \Pr[f_{\Theta,S_n}(\boldsymbol{x}) \neq y \big| |C_i \cap S_n| > \tau, \boldsymbol{x} \in C_i] \Pr[|C_i \cap S_n| > \tau, \boldsymbol{x} \in C_i] \ .$$

From $C_i \in \Lambda_3$, all training examples in $C_i$ have the same label, and we assume positive training examples in $C_i$ without loss of generality. Then, we have $f_{\Theta,S_n}(\boldsymbol{x}) = 1$ for all $\boldsymbol{x} \in C_i$. Denote by the expected conditional probability over cell $C_i$

$$\bar{\eta}(C_i) = E[\eta(\boldsymbol{x}) | \boldsymbol{x} \in C_i] \ .$$

If $\bar{\eta}(C_i) \geq 1 - \epsilon$, then we have

$$\Pr[f_{\Theta,S_n}(\boldsymbol{x}) \neq y \big| |C_i \cap S_n| > \tau, \boldsymbol{x} \in C_i] \Pr[|C_i \cap S_n| > \tau, \boldsymbol{x} \in C_i] \leq \epsilon \ ;$$

If $\bar{\eta}(C_i) < 1 - \epsilon$ and $C_i \in \Lambda_3$, then we have

$$\Pr[f_{\Theta,S_n}(\boldsymbol{x}) \neq y \big| |C_i \cap S_n| > \tau, \boldsymbol{x} \in C_i] \Pr[|C_i \cap S_n| > \tau, \boldsymbol{x} \in C_i]$$
$$\leq \ \Pr[|C_i \cap S_n| > \tau] \leq \exp(-\tau\epsilon) \ .$$

It follows that, for $C_i \in \Lambda_3$,

$$\Pr[C_i] \Pr\left[f_{\Theta,S_n}(\boldsymbol{x}) \neq y \Big| \boldsymbol{x} \in C_i\right] \leq \Pr[C_i] \Pr[|C_i \cap S_n| \leq \tau] + \Pr[C_i](\epsilon + \exp(-\tau\epsilon)) \ .$$

By setting $\epsilon = (\ln \tau)/\tau$, we have

$$\Pr[C_i] \Pr\left[f_{\Theta,S_n}(\boldsymbol{x}) \neq y \Big| \boldsymbol{x} \in C_i\right] \leq \Pr[C_i] \Pr[|C_i \cap S_n| \leq \tau] + \Pr[C_i] \frac{1 + \ln \tau}{\tau} \ .$$

It follows that, by combining with Lemma 14 and Eqns. (32)-(33)

$$E_{S_n,\Theta}[R_{\mathcal{D}}(f_{S_n,\Theta})] \leq \frac{k\tau}{n}\left(1 + \sqrt{\frac{2}{\tau}}\right) + \frac{1}{\tau}(1 + \ln \tau)$$

$$+ \sum_{i=1}^{k} E_{S_n,\Theta}\left[\Pr[f_{\Theta,S_n}(\boldsymbol{x}) \neq y | \boldsymbol{x} \in C_i] \Pr[\boldsymbol{x} \in C_i] \mathbb{I}[C_i \notin \Lambda_3]\right] \ . \tag{34}$$

For $C_i \notin \Lambda_3$, we have different labels in $C_i$. It follows the height $h(C_i) \geq \log_2 k - 2$ and the splitting times for each dimension are more than $(\log_2 k - 2)/d - 1$ from the construction of random tree in Algorithm 1. Hence, we upper bound the diameter of rectangle cell $C_i$ as follows:

$$\nu(C_i) \leq \sqrt{d}\left(\frac{1}{2}\right)^{(\log_2 k - 2)/d - 1} = \frac{2^{1+2/d}\sqrt{d}}{k^{1/d}} \leq \frac{8\sqrt{d}}{k^{1/d}} \ .$$

It follows that, from Lemma 4 and Eqn. (34),

$$E_{S_n,\Theta}[R_{\mathcal{D}}(f_{S_n,\Theta})]$$

$$\leq \ R_{\mathcal{D}}^* + \frac{k\tau}{n}\left(1 + \sqrt{\frac{2}{\tau}}\right) + \frac{1}{\tau}(1 + \ln \tau) + \frac{2L\sqrt{d}}{k^{1/d}} + \frac{3k}{n} + \sum_{i=1}^{k}\sqrt{\frac{\Pr[C_i]}{n}}$$

$$\leq \ R_{\mathcal{D}}^* + \frac{k\tau}{n}\left(1 + \sqrt{\frac{2}{\tau}}\right) + \frac{1}{\tau}(1 + \ln \tau) + \frac{2L\sqrt{d}}{k^{1/d}} + \frac{3k}{n} + \sqrt{\frac{k}{n}} \ .$$

We have, by setting $\tau = \left\lceil \sqrt{n \ln n / k} \right\rceil$ and algebra calculations,

$$E_{S_n, \Theta}[R_{\mathcal{D}}(f_{S_n, \Theta})]$$

$$\leq R_{\mathcal{D}}^* + \frac{k\tau}{n} + \frac{\sqrt{2\tau}k}{n} + \frac{1}{\tau}(1 + \ln\tau) + \frac{2L\sqrt{d}}{k^{1/d}} + \frac{3k}{n} + \sqrt{\frac{k}{n}}$$

$$\leq R_{\mathcal{D}}^* + \sqrt{\frac{k \ln n}{n}} + \sqrt[4]{\frac{4k^3 \ln n}{n^3}} + \sqrt{\frac{k}{n \ln n}} \left(1 + \frac{1}{2}\ln\frac{n \ln n}{k}\right) + \frac{6k}{n} + \sqrt{\frac{k}{n}} + \frac{2L\sqrt{d}}{k^{1/d}}$$

$$\leq R_{\mathcal{D}}^* + \sqrt{\frac{k \ln n}{n}} + \sqrt[4]{\frac{4k^3 \ln n}{n^3}} + \sqrt{\frac{k}{n \ln n}} \left(1 + \frac{1}{2}\ln\frac{n \ln n}{k}\right) + \frac{6k}{n} + \sqrt{\frac{k}{n}} + \frac{2L\sqrt{d}}{k^{1/d}}$$

$$\leq R_{\mathcal{D}}^* + 2\sqrt{\frac{k \ln n}{n}} + \sqrt[4]{\frac{4k^3 \ln n}{n^3}} + \frac{6k}{n} + 2\sqrt{\frac{k}{n}} + \frac{2L\sqrt{d}}{k^{1/d}} \quad (n \geq 4, k \geq 2),$$

which completes the proof by combining with Lemma 1. ∎

## G  Proof of Theorem 4

The proof is essentially similar to that of Theorems 3. Given a random tree classifier $f_{\Theta, S_n}(\boldsymbol{x})$ with $k$ leaves ($k \leq k_0$), let $C_1, C_2, \ldots, C_k$ be a partition of the instance space $\mathcal{X}$. Based on the law of total probability, we have the classification error of random forests classifier $f_{\Theta, S_n}(\boldsymbol{x})$ over distribution $\mathcal{D}$

$$R_{\mathcal{D}}(f_{S_n, \Theta}) = \Pr_{(\boldsymbol{x}, y) \sim \mathcal{D}}[f_{\Theta, S_n}(\boldsymbol{x}) \neq y]$$

$$= \sum_{i=1}^{k} \Pr[f_{\Theta, S_n}(\boldsymbol{x}) \neq y | \boldsymbol{x} \in C_i] \Pr[\boldsymbol{x} \in C_i]. \tag{35}$$

For any $i \in [k]$ and $\tau \geq 1$, we have

$$\Pr[C_i] \Pr\left[f_{\Theta, S_n}(\boldsymbol{x}) \neq y \Big| \boldsymbol{x} \in C_i\right]$$

$$= \Pr[C_i] \Pr[f_{\Theta, S_n}(\boldsymbol{x}) \neq y, |C_i \cap S_n| \leq \tau | \boldsymbol{x} \in C_i] \tag{36}$$

$$+ \Pr[C_i] \Pr[f_{\Theta, S_n}(\boldsymbol{x}) \neq y, |C_i \cap S_n| > \tau | \boldsymbol{x} \in C_i]. \tag{37}$$

From Lemma 14, we have

$$E_{S_n}[\Pr[C_i] \Pr[f_{\Theta, S_n}(\boldsymbol{x}) \neq y, |C_i \cap S_n| \leq \tau | \boldsymbol{x} \in C_i]] \leq \frac{\tau}{n} \left(1 + \sqrt{\frac{2}{\tau}}\right). \tag{38}$$

From the assumption in Theorem 4, we see that all training examples in each $C_i$ have the same label, and we assume positive training examples in $C_i$ without loss of generality. It follows that $f_{\Theta, S_n}(\boldsymbol{x}) = 1$ for all $\boldsymbol{x} \in C_i$. Denote by

$$\bar{\eta}(C_i) = E[\eta(\boldsymbol{x}) | \boldsymbol{x} \in C_i]$$

the expected conditional probability over the rectangle cell $C_i$. If $\bar{\eta}(C_i) \geq 1 - \epsilon$, then we have

$$\Pr[f_{\Theta, S_n}(\boldsymbol{x}) \neq y, |C_i \cap S_n| > \tau | \boldsymbol{x} \in C_i] \leq \epsilon;$$

If $\bar{\eta}(C_i) < 1 - \epsilon$ and $C_i \in \Lambda_3$, then we have

$$\Pr[f_{\Theta, S_n}(\boldsymbol{x}) \neq y, |C_i \cap S_n| > \tau | \boldsymbol{x} \in C_i] \leq (1 - \epsilon)^\tau \leq \exp(-\tau\epsilon).$$

It follows that, by setting $\epsilon = (\ln\tau)/\tau$,

$$\Pr[f_{\Theta, S_n}(\boldsymbol{x}) \neq y, |C_i \cap S_n| > \tau | \boldsymbol{x} \in C_i] \leq \epsilon + \exp(-\tau\epsilon) \leq (1 + \ln\tau)/\tau. \tag{39}$$

Combining with Eqns. (35)-(39), we have

$$E_{S_n, \Theta}[R_{\mathcal{D}}(f_{S_n, \Theta})] \leq \frac{k\tau}{n}\left(1 + \sqrt{\frac{2}{\tau}}\right) + \frac{1}{\tau}(1 + \ln\tau)$$

$$\leq \frac{k_0 \tau}{n}\left(1 + \sqrt{\frac{2}{\tau}}\right) + \frac{1}{\tau}(1 + \ln\tau).$$

By setting $\tau = \left\lceil \sqrt{n \ln n / k_0} \right\rceil$ and simple algebraic calculations, we have

$$E_{S_n, \Theta}[R_{\mathcal{D}}(f_{S_n, \Theta})] \le 2\sqrt{\frac{k_0 \ln n}{n}} + \sqrt[4]{\frac{4k_0^3 \ln n}{n^3}} + \frac{3k_0}{n} + \sqrt{\frac{k_0}{n \ln n}} \,,$$

which completes the proof by combining with Lemma 1. ∎