[Reviews · NeurIPS 2020]

Review 1

Summary and Contributions: The paper discusses classification with random forests and gives convergence rates to the Bayes error for various methods to construct the individual trees: pure random trees with random splits of the leaf-cells, random trees with midpoint splits, and a simplified version of Breimans random forest, where splitting terminates on cells which only contain example points of equal labels. In the latter case the rate matches the minimax rates for plug-in classifiers up to a logarithmic factor. This is further improved under an assumption on the data distribution. All rates are in expectation.

Strengths: I did not study the entire appendix, but I checked the proofs in the main part of the paper and of several supporting lemmata in the appendix. They seemed correct (but see caveat below). I liked the estimate on the height of a random tree. I am not an expert on random forests, but the reviewing duties motivated a brief survey of the literature, where I did not find comparable results on classification. As random forests are widely used in practice, their theoretical study seems quite relevant.

Weaknesses: There are several ambiguities and imprecisions. I very much disliked the argument in the proof of Theorem 1 where the unit cube in R^d is treated as a discrete set by summing over its elements, although I am convinced that the argument can be made correct without damage to the result. This kind of hand-waving would be more acceptable if there was a rigorous proof in the appendix. The assumption on the distribution in Theorem 4 is in terms of the algorithm itself and remains quite obscure. Some elucidation of the assumption would be in order. There are a number of small grammatical errors, mostly missing articles or near typos like "3D objection recognition". The paper would benefit from a proof-reading in this respect.(selecting "a dimension at random, uniformly over dimensions of the longest side length" presumably means random selection of any dimension along which the side length of the cell is maximal). ---------------------------update-------------------------------------------------- The authors have addressed these concerns and promise to make corresponing improvements. In the hope that they keep this promise I improve my score to 7.

Correctness: The claims seem correct, but please see the previous comments.

Clarity: Apart from the many small grammatical errors, the paper is well written and various methods of tree construction are well explained.

Relation to Prior Work: This is done in the introduction and in the section on related work on page 6. As mentioned above I am not an expert on random forests, and the discussion in the paper may not be completely exhaustive.

Reproducibility: Yes

Additional Feedback: Please check the grammar in another reading. Otherwise see the comments above.


Review 2

Summary and Contributions: This work studies the finite sample convergence of random forests when the probability of class one is Lipschitz. Rates are derived of the form n^{-1/cd}, which scale extremely poorly in dimension but are somewhat intuitive. When the data is sufficiently structured the scaling is improved to the parametric rate.

Strengths: See above.

Weaknesses: No experiments, which could be helpful to verify the derived rates/identify if they are pessimistic.

Correctness: The claims seem technically correct.

Clarity: Clearly written. Minor points: First sentence of the conclusion and introduction are nearly identical, which is somewhat jarring. Line 181: Intuitive -> Intuitively Line 182: makes quite -> makes it quite Lines 56-57: missing "the" several places. Line 57: estimation -> estimate Line 35: when do random -> when random Line 34: along variable -> along with variable

Relation to Prior Work: Yes.

Reproducibility: Yes

Additional Feedback:


Review 3

Summary and Contributions: This article deals with the generalization performance of random forests. The authors establish convergence rates (non-asymptotic bounds) for different algorithms used for pattern classification (computation of dichotomies). They introduce a simplified variant of Breiman's original random forest (Algorithm 1). For this classifier, under mild hypotheses, the convergence rate is optimal up to a logarithmic factor.

Strengths: The theory of random forests has primarily been developed for regression, not pattern classification. Thus, this paper contributes to bridging the gap between theory and practice. It is noticeable that the formula of Theorem 4 does not depend on the dimension d of the description space.

Weaknesses: There is no argument suggesting that those theorems could extend nicely to the multi-class case. I doubt that it could be the case.

Correctness: I checked large parts of the computations without identifying any flaw.

Clarity: The paper is clearly written. However, the scope of the results obtained could be discussed in more detail.

Relation to Prior Work: The reader unfamiliar with the domain will find it difficult to figure out what the state of the art is (for pattern classification).

Reproducibility: Yes

Additional Feedback:


Review 4

Summary and Contributions: The paper provides finite-sample convergence rates for two simplified variants of random forests: - Breiman's pure random forests, where k nodes are split at random dimensions and at (uniformly) random positions or at midpoint. The latter giving faster convergence rates than the former. - A new variant where nodes are split at midpoint along one of their longest dimensions picked at random and where node splitting is stopped as soon as all examples in a node are of the same class. Better convergence rates are also derived for this latter method by making further assumption about the problem.

Strengths: - While convergence of simplified RF models has been studied in regression, this paper proposes the first convergence rate analyses in the context of classification. Results are non trivial and they also nicely follows intuition, in the sense that the faster methods are the expected ones. - The second simplified method that is studied is closer to the original random forests than simplified RF methods usually analysed in the literature. - Convergence rates come with interesting new side results, such as a new link between the convergence of forests and trees and a better estimation of the height of random trees.

Weaknesses: - The studied algorithms remain quite far from real random forests (no bootstrap sampling, split choices are fully independent of the data, trees are pruned, etc.) - As in other results in the literature, convergence rates for forests are by-product of convergence rate of individual trees (using Lemma 1). The results therefore do not really show the benefit of using forests instead of trees in terms of convergence rate. This should be discussed in the paper I think. - Overall, the contribution is purely theoretical. No real conclusion is drawn from the theoretical results that would help better understand standard RF or suggest modification to these methods. - The structure of the paper could be improved, as well as the discussion of related works (see below). - The paper is very technical with 14 pages of mathematical proofs in the supplement and 2.5 additional pages of proofs of Theorem 1 in the main text. I think this kind of very technical contribution would be more appropriate for a journal submission than for a conference (given the limited time allotted for reviewing). ---------- update after the author's response ---------- I thank the authors for their response. They mostly confirm the limitations that I highlighted in my review, which do not prevent acceptance. Concerning A3, the theoretical results indeed allow to compare simplified RF models but they do not really motivate changes to the standard RF algorithm (which uses neither random splits, nor midpoint splits for example).

Correctness: The results are plausible to me. However, I didn't fully check the proofs in the supplement.

Clarity: The paper is clearly written. One problem that I see is however in the definition of the random variables Y_i and U_i between lines 254 and 261. These variables are related to a specific dimension j. They should thus be indexed by j in addition to i. Note that Lemma 3 does not need them. These variables are only needed for the proof of Lemma 8 in Appendix C that considers a fixed dimension j. There are several redundancies in the text, with several statements repeated several times between the introduction, related works and conclusions. I think the paper could also be better structured. All main results are gathered in Section 4 in a very unstructured way. I think this section could be subdivided into several sections corresponding to the different simplified RF models and the related works could be also addressed in a separate section.

Relation to Prior Work: The discussion of related works could be improved. Between lines 198 and 207, the authors discuss several random forests extensions that I think are unrelated to the present contribution. It would have been interesting to discuss instead in more details convergence rates derived in the literature for simplified RF models in regression (e.g., in [8] or [25]).

Reproducibility: Yes

Additional Feedback: Some typos: - Line 181: Intuitive => Intuitively - Lines 236-237: "the node of containing x" => "the node containing x" - Line 222 and also line 298: "it is interesting to exploit the convergence rates": Do you mean "explore" or "analyse"?

[Author Response · NeurIPS 2020]

We want to thank reviewers for insightful comments, and we will improve the paper accordingly. In the following we
focus on technical questions.

**[To reviewer 1]**

**[Q1]** ... the argument in the proof of Theorem 1 where the unit cube in R^d is treated as a discrete set by summing over
its elements ... This kind of hand-waving would be more acceptable if there was a rigorous proof in the appendix.
**[A1]** We will clarify that Theorem 1 holds for discrete and continuous sets in R^d, while we use the summing over its
elements in the proof of Theorem 1 for simplicity and readers' understanding. We will present the detailed proof in the
appendix based on expectation and probability, rather than the summing over its elements, which is suitable for discrete
and continuous distributions.

**[Q2]** The assumption on the distribution in Theorem 4 is in terms of the algorithm itself and remains quite obscure.
Some elucidation of the assumption would be in order.
**[A2]** We will clarify that the assumption in Theorem 4 is relevant to algorithm, and it also holds for some irrelevant
cases, for example, Algorithm 1 in our work satisfies such assumption when the data is separable and the separable
hyperplane is parallel to axis.

We will improve this work according to your suggestions. Thank you.

**[To Reviewer 2]**

**[Q1]** No experiments, which could be helpful to verify the derived rates/identify if they are pessimistic.
**[A1]** We will clarify that this is a theory paper and the main arguments are supported by proofs that are rigorous and
more reliable (experiments may be less reliable because many factors, such as parameter tuning and sampling, may
influence the result). In a future longer version, we will consider the design of experiments.

We will improve this work according to your suggestions. Thank you.

**[To Reviewer 3]**

**[Q1]** There is no argument suggesting that those theorems could extend nicely to the multi-class case.
**[A1]** We will clarify that this work focuses on binary classification, and it is interesting to extend our work to multi-class
learning, where the challenges lie in the theoretical analysis of predictions $f(x, y) - \max_{i \neq y} f(x, i)$ and Lipshcitz
assumptions over multiple class-conditional distributions. Chen & Sun (JMLR 2016) and Ramaswamy et al. (ArXiv
2015) may shed some lights on this direction.

We will improve this work according to your suggestions. Thank you.

**[To Reviewer 4]**

**[Q1]** The studied algorithms remain quite far from real random forests (no bootstrap sampling, split choices are fully
independent of the data, trees are pruned, etc.)
**[A1]** We will clarify that this work takes one step towards convergence rate of random forest for classification, and
Algorithm 1 follows Breiman's random forests but with different splitting dimension and position. We will also clarify
that it is still a long way to fully understand random forests and relevant mechanisms such as bootstrap sampling,
data-dependence tree structure, tree pruning, etc. We leave those to future work.

**[Q2]** As in other results in the literature, convergence rates for forests are by-product of convergence rate of individual
trees (using Lemma 1) ... This should be discussed in the paper I think.
**[A2]** We will clarify that the convergence rates of random forests are obtained from the expectation of convergence
rates of individual trees based on Lemma 1, which can be viewed as the average of convergence rates of all of individual
random trees.

**[Q3]** No real conclusion is drawn from the theoretical results that would help better understand standard RF or suggest
modification to these methods.
**[A3]** We will clarify that our work is beneficial to understand the splitting mechanisms for random forests, such as the
selections of splitting leaves, dimensions and positions, which may motivate new methods. For example, we get better
convergence rates for pure random forests by using midpoint splits than random split (Thms 1 and 2); we also get better
convergence rates by considering different selections of splitting leaves and dimensions in Algorithm 1 (Thms 3 and 4).

**[Q4]** One problem ... definition of the random variables $Y_i$ and $U_i$ between lines 254 and 261 ... They should thus be
indexed by $j$ in addition to $i$ ... needed for the proof of Lemma 8 in Appendix C that considers a fixed dimension $j$.
**[A4]** We will add indexer $i$ and $j$ to random variables $Y$ and $U$, and move relevant definitions before Lemma 8.

We will reorganize this work, add more discussions, and improve this work according to your suggestions. Thank you.

[Meta-Review · NeurIPS 2020]

The paper provides finite-sample convergence rates for two simplified variants of random forests. Overall, the contribution is purely theoretical. I personally think that this work shed new interesting ideas on the behavior of a learning algorithm that is intensively used world wide. This work clearly deserve a poster acceptation at NeurIPS.